# ψDAG: Projected Stochastic Approximation Iteration for DAG Structure Learning

## Abstract

Learning the structure of Directed Acyclic Graphs (DAGs) presents a significant challenge due to the vast combinatorial search space of possible graphs, which scales exponentially with the number of nodes. Recent advancements have redefined this problem as a continuous optimization task by incorporating differentiable acyclicity constraints. These methods commonly rely on algebraic characterizations of DAGs, such as matrix exponentials, to enable the use of gradient-based optimization techniques. Despite these innovations, existing methods often face optimization difficulties due to the highly non-convex nature of DAG constraints and the per-iteration computational complexity. In this work, we present a novel framework for learning DAGs, employing a Stochastic Approximation approach integrated with Stochastic Gradient Descent (SGD)-based optimization techniques. Our framework introduces new projection methods tailored to efficiently enforce DAG constraints, ensuring that the algorithm converges to a feasible local minimum. With its low iteration complexity, the proposed method is well-suited for handling large-scale problems with improved computational efficiency. We demonstrate the effectiveness and scalability of our framework through comprehensive experimental evaluations, which confirm its superior performance across various settings.

## 1 Introduction

Learning graphical structures from data using Directed Acyclic Graphs (DAGs) is a fundamental challenge in machine learning (Koller & Friedman, 2009; Peters et al., 2016; Arjovsky et al., 2019; Sauer & Geiger, 2021). This task has a wide range of practical applications across fields such as economics, genome research (Zhang et al., 2013; Stephens & Balding, 2009), social sciences (Morgan & Winship, 2015), biology (Sachs et al., 2005a), and causal inference (Pearl, 2009; Spirtes et al., 2000). Learning the graphical structure is essential because the resulting models can often be given causal interpretations or transformed into representations with causal significance, such as Markov equivalence classes. When graphical models cannot be interpreted causally (Pearl, 2009; Spirtes et al., 2000), they can still offer a compact and flexible representation for decomposing the joint distribution.

Structure learning methods are typically categorized into two approaches: score-based algorithms searching for a DAG minimizing a particular loss function and constraint-based algorithms relying on conditional independence tests. Constraint-based methods, such as the PC algorithm (Spirtes & Glymour, 1991) and FCI (Spirtes et al., 1995; Colombo et al., 2012), use conditional independence tests to recover the Markov equivalence class under the assumption of faithfulness. Other approaches, like those described in Margaritis & Thrun (1999) and Tsamardinos et al. (2003), employ local Markov boundary search. On the other hand, score-based methods frame the problem as an optimization of a specific scoring function, with typical choices including BGe (Kuipers et al., 2014), BIC (Chickering & Heckerman, 1997), BDe(u) (Heckerman et al., 1995), and MDL (Bouckaert, 1993). Given the vast search space of potential graphs, many score-based methods employ local heuristics, such as Greedy Equivalence Search (GES) (Chickering, 2002), to efficiently navigate this complexity. Additionally, Tsamardinos et al. (2006), Gámez et al. (2011) propose hybrid methods combining elements of both constraint-based and score-based learning.

Recently, Zheng et al. (2018) introduced a smooth formulation for enforcing acyclicity, transforming the structure learning problem from its inherently discrete nature into a continuous, non-convex optimization task. This formulation allows for the use of gradient-based optimization techniques, enabling various extensions and adaptations to various domains, including nonlinear models (Yu et al., 2019; Ng et al., 2022b; Kalainathan et al., 2022), interventional datasets (Brouillard et al., 2020; Faria et al., 2022), unobserved confounders (Bhattacharya et al., 2021; Bellot & Van der Schaar, 2021), incomplete datasets (Gao et al., 2022a; Wang et al., 2020), time series analysis (Sun et al., 2021; Pamfil et al., 2020), multi-task learning (Chen et al., 2021), multi-domain settings (Zeng et al., 2021), federated learning (Ng & Zhang, 2022; Gao et al., 2023), and representation learning (Yang et al., 2021). With the growing interest in continuous structure learning methods (Vowels et al., 2022), a variety of theoretical and empirical studies have emerged. For instance, Ng et al. (2020) investigated the optimality conditions and convergence properties of continuously constrained approaches such as Zheng et al. (2018). In the bivariate case, Deng et al. (2023b) demonstrated that a suitable optimization strategy converges to the global minimum of the least squares objective. Additionally, Zhang et al. (2022) and Bello et al. (2022) identified potential gradient vanishing issues with existing DAG constraints (Zheng et al., 2018) and proposed adjustments to overcome these challenges.

**Contributions.**    In this work, we focus on the graphical models represented as Directed Acyclic Graphs (DAGs). Our main contributions can be summarized as follows:

1. **Problem reformulation:** We introduce a new reformulation (8) of the discrete optimization problem for finding DAG as a stochastic optimization problem and we discuss its properties in detail in Section 3.1. We demonstrate that the solution of this reformulated problem recovers the true DAG (Section 3.1).

2. **Novel algorithm:** Leveraging insights from stochastic optimization, we present a new framework (Algorithm 1) for DAG learning (Section 4) and present a simple yet effective algorithm $\psi$DAG (Algorithm 3) within the framework. We proved that Algorithm $\psi$DAG converges to a local minimum of problem (8).

3. **Experimental comparison:** In Section 5, we demonstrate that the method $\psi$DAG scales very well with graph size, handling up to 10000 nodes. At that scale, the primary limitation is not computation complexity but the memory required to store the DAG itself. As a baseline, we compare $\psi$DAG with established DAG learning methods, including NOTEARS (Zheng et al., 2018), GOLEM (Ng et al., 2020) and DAGMA (Bello et al., 2022). We show a significant improvement in scalability, as baseline methods struggle with larger graphs. Specifically, NOTEARS (Zheng et al., 2018), GOLEM (Ng et al., 2020) and DAGMA (Bello et al., 2022) require more than 100 hours for graphs with over 3000 nodes, exceeding the allotted time.

## 2 BACKGROUND

### 2.1 GRAPH NOTATION

Before discussing the connection to the most relevant literature, we formalize the graph notation associated with DAGs.

Let $\mathcal{G} \stackrel{def}{=} (V, E, w)$ represent a weighted directed graph, where $V$ denotes the set of vertices with cardinality $d \stackrel{def}{=} |V|$, $E \in 2^{V \times V}$ is the set of edges, and $w : V \times V \to \mathbb{R} \setminus \{0\}$ assigns weights to the edges. The *adjacency matrix* $\mathbf{A}(\mathcal{G}) : \mathbb{R}^{d \times d}$ is defined such that $[\mathbf{A}(\mathcal{G})]_{ij} = 1$ if $(i, j) \in E$ and $0$ otherwise. Similarly, the *weighted adjacency matrix* $\mathbf{W}(\mathcal{G})$ is defined by $[\mathbf{W}(\mathcal{G})]_{ij} = w(i, j)$ if $(i, j) \in E$ and $0$ otherwise.

When the weight function $w$ is binary, we simplify the notation to $\mathcal{G} \stackrel{def}{=} (V, E)$. Similarly, when the graph $\mathcal{G}$ is clear from context, we shorthand the notation to $\mathbf{A} \stackrel{def}{=} \mathbf{A}(\mathcal{G})$ and $\mathbf{W} \stackrel{def}{=} \mathbf{W}(\mathcal{G})$.

We denote the space of DAGs as $\mathbb{D}$. Since we will be utilizing topological sorting of DAGs[1], we also denote the space of vertex permutations $\Pi$.

## 2.2 LINEAR DAG AND SEM

A Directed Acyclic Graph (DAG) model, defined on a set of $n$ random vectors $\mathbf{X} \in \mathbb{R}^{n \times d}$, where $\mathbf{X} \stackrel{def}{=} (X_1, \ldots, X_n)$ and $X_i \in \mathbb{R}^d$, consists of two components:

1. A DAG $\mathcal{G} = (V, E)$, which encodes a set of conditional independence relationships among the variables.

2. The joint distribution $P(\mathbf{X})$ with density $p(x)$, which is Markov with respect to the DAG $\mathcal{G}$ and factors as $p(x) = \prod_{i=1}^{d} p(x_i \mid x_{\text{PA}_{\mathcal{G}}(i)})$, where $\text{PA}_{\mathcal{G}}(i) = \{j \in V : X_j \to X_i \in E\}$ represents the set of parents of $X_i$ in $\mathcal{G}$.

This work focuses on the linear DAG model, which can be equivalently represented by a set of linear Structural Equation Models (SEMs). In matrix notation, the linear DAG model can be expressed as

$$\mathbf{X} = \mathbf{X}\mathbf{W} + \mathbf{N}, \tag{1}$$

where $\mathbf{W} = [\mathbf{W}_1 | \cdots | \mathbf{W}_d]$ is a weighted adjacency matrix, and $\mathbf{N} \stackrel{def}{=} (N_1, \ldots, N_n)$ is a matrix where each $N_i \in \mathbb{R}^d$ represents a noise vector with independent components. The structure of graph $\mathcal{G}$ is determined by the non-zero coefficients in $\mathbf{W}$; specifically $X_j \to X_i \in E$ if and only if the corresponding coefficient in $\mathbf{W}_i$ for $X_j$ is non-zero. The classical objective function is based on the least squares loss applied to the linear DAG model,

$$l(\mathbf{W}; \mathbf{X}) \stackrel{def}{=} \frac{1}{2n} \|\mathbf{X} - \mathbf{X}\mathbf{W}\|_F^2. \tag{2}$$

## 2.3 MOST RELATED LITERATURE

A significant body of research in DAG learning revolves around non-convex continuous optimization frameworks, such as NOTEARS (Zheng et al., 2018), GOLEM (Ng et al., 2020), and DAGMA (Bello et al., 2022). These approaches address the DAG constraint using either smooth approximations or novel penalty functions, but they are often computationally expensive and lack scalability.

Zheng et al. (2018) formulated the DAG learning problem as a constrained optimization task, minimizing the least squares loss while enforcing acyclicity through the matrix exponential. While this method achieves state-of-the-art results for smaller graphs, its cubic complexity for computing the acyclicity term severely limits its scalability. Ng et al. (2020) enhanced the scoring function by incorporating a log-determinant term aligned with the Gaussian likelihood, which improves efficiency but does not guarantee acyclic solutions. Similarly, Bello et al. (2022) introduced a differentiable and exact log-determinant-based acyclicity constraint, but its reliance on augmented Lagrangian methods introduces hyperparameter tuning challenges and potential numerical instability.

Other works, such as Chen et al. (2019), proposed variance-ordering procedures for estimating topological orderings under equal error variances. While these methods naturally extend to high-dimensional settings, their reliance on controlling the maximum in-degree of the graph becomes computationally intensive as graph density increases. In contrast, $\psi$DAG avoids these assumptions and demonstrates scalability to graphs with up to $10,000$ nodes. Gao et al. (2022b) focused on theoretical guarantees for Gaussian DAG models, deriving minimax optimal bounds for structure recovery. Although their work offers valuable insights into sample efficiency, it does not address the computational challenges of large-scale DAG learning. Our approach complements this by providing a scalable stochastic optimization framework applicable to broader settings.

Wei et al. (2020) examined optimization challenges in NOTEARS by analyzing KKT conditions and proposed the KKTS algorithm as a post-processing enhancement. While this method improves structural Hamming distance (SHD), its reliance on specific constraints and post-hoc refinements

---

[1]Topologial sorting of a graph $\mathcal{G} \stackrel{def}{=} (V, E, w)$ refers to vertex ordering $V_1, V_2, \ldots, V_d$ such that $E$ contains no edges of the form $V_i \to V_j$, where $i \leq j$. Importantly, every DAG has at least one topological sorting.

limits its applicability. By contrast, $\psi\mathsf{DAG}$ reformulates DAG learning as a stochastic optimization problem, seamlessly integrating gradient-based methods for large-scale graphs.

Additionally, Deng et al. (2023a) introduced a bi-level algorithm that iteratively refines topological orders via node swaps, achieving local minima or KKT points. However, this approach is constrained by a specific function $h(B) = \sum_{i=1}^{d} c_i \mathrm{Tr}(B^i)$, which is computationally expensive and limits its scalability to applications involving larger graphs. Consequently, their experiments are restricted to synthetic datasets with graphs containing up to $d = 100$ nodes. Moreover, the algorithm initializes the $\mathbf{W}$ matrix using linear regression coefficients in the least squares case, resulting in a different starting point for optimization, which makes direct comparisons with other methods challenging. Our method addresses these limitations by generalizing the DAG learning framework and demonstrating superior scalability and performance on both synthetic and real datasets.

While many of these works focus on specific assumptions, penalty terms, or theoretical guarantees, our framework prioritizes scalability, flexibility, and applicability. The integration of stochastic optimization enables $\psi\mathsf{DAG}$ to tackle large graphs effectively, establishing it as a robust and practical solution for DAG learning challenges. For additional details of related work, see Appendix A.

## 3 STOCHASTIC APPROXIMATION FOR DAGs

Our framework is built on a reformulation of the objective function as a stochastic optimization problem, aiming to minimize the stochastic function $F(w)$,

$$\min_{w \in \mathbb{R}^d} \left\{ F(w) \stackrel{def}{=} \mathbb{E}_{\xi} \left[ f(w, \xi) \right] \right\}, \tag{3}$$

where $\xi \in \Xi$ is a random variable that follows the distribution $\Xi$. This formulation is common in stochastic optimization where computing the exact expectation is infeasible, but the values of $f(w, \xi)$ and its stochastic gradients $g(w, \xi)$ can be computed. Linear and logistic regressions are classical examples of such problems.

To address this problem, two main approaches exist: Stochastic Approximation (SA) and Sample Average Approximation (SAA). The SAA approach involves sampling a fixed number $n$ of random variables or data points $\xi_i$ and then minimizing their average $\tilde{F}(w)$:

$$\min_{w \in \mathbb{R}^d} \left\{ \tilde{F}(w) \stackrel{def}{=} \frac{1}{n} \sum_{i=1}^{n} f(w, \xi_i) \right\}. \tag{4}$$

Now, the problem (4) becomes deterministic and can be solved using various optimization methods, such as gradient descent. However, the main drawback of this approach is that the solution of (4) $\tilde{w}^*$ is not necessarily equal to the solution of the original problem (3). Even with a perfect solution of (4), there will still be a gap $\|\tilde{w}^* - w^*\| = \delta_x$ and $F(\tilde{w}^*) - F^* = \delta_F$ between approximate and true solution. These gaps are dependent on the sample size $n$.

Stochastic Approximation (SA) minimizes the true function $F(w)$ by utilizing the stochastic gradient $g(w, \xi)$. Below, we provide the formal definition of a stochastic gradient.

**Assumption 1.** *For all $w \in \mathbb{R}^d$, we assume that stochastic gradients $g(w, \xi) \in \mathbb{R}^d$ satisfy*

$$\mathbb{E}[g(w, \xi) \mid w] = \nabla F(w), \quad \mathbb{E} \left[ \|g(w, \xi) - \nabla F(w)\|^2 \mid w \right] \leq \sigma_1^2. \tag{5}$$

We use these stochastic gradients in $\mathsf{SGD}$-type methods:

$$w_{t+1} = w_t - h_t g(w_t, \xi_i), \tag{6}$$

where $h_t$ is a step-size schedule. SA originated with the pioneering paper by Robbins & Monro (1951). For convex and $L$-smooth function $F(w)$, Polyak (1990); Polyak & Juditsky (1992); Nemirovski et al. (2009); Nemirovski & Yudin (1983) developed significant improvements to SA method in the form of longer step-sizes with iterate averaging, and obtained the convergence guarantee

$$\mathbb{E}\left[F(w_T) - F(x^*)\right] \leq \mathcal{O}\left(\frac{\sigma_1 R}{\sqrt{T}} + \frac{L_1 R^2}{T}\right).$$

Lan (2012) developed an optimal method with a guaranteed convergence rate $\mathcal{O}\left(\frac{\sigma_1 R}{\sqrt{T}} + \frac{L_1 R^2}{T^2}\right)$, matching the worst-case lower bounds. The key advantage of SA is that it provides convergence guarantees for the original problem (3). Additionally, methods effective for the SA approach tend to perform well for the SAA approach as well.

## 3.1 STOCHASTIC REFORMULATION

Using the perspective of Stochastic Approximation, we can rewrite the linear DAG (1) as

$$x = X_i = \left[\mathbf{I} - \mathbf{W}_*^\top\right]^{-1} N_i, \tag{7}$$

where $\mathbf{W}^*$ is a true DAG that corresponds to the full distribution, and our goal is to find DAG $\mathbf{W}$ that is close to $\mathbf{W}^*$. If we assume that $x = X_i$ is a random vector sampled from a distribution $\mathcal{D}$, we can express the objective function as an expectation,

$$\min_{\mathbf{W} \in \mathbb{D}} \mathbb{E}_{x \sim \mathcal{D}} \left[ l(\mathbf{W}; x) \stackrel{def}{=} \tfrac{1}{2}\|x - \mathbf{W}^\top x\|^2 = \tfrac{1}{2}\|x^\top - x^\top \mathbf{W}\|^2 \right]. \tag{8}$$

For $x$ from (7) we can calculate $\|x - \mathbf{W}^\top x\| = \|(\mathbf{I} - \mathbf{W}^\top)x\| = \|(\mathbf{I} - \mathbf{W})\left[\mathbf{I} - \mathbf{W}_*^\top\right]^{-1} N_i\|$, which implies that the minimizer of (8) recovers the true DAG. Conversely, this is not the case for methods such as Zheng et al. (2018), Ng et al. (2020), and Bello et al. (2022), which are based on SAA approaches with losses (2), (10), (11), (12).

# 4 SCALABLE FRAMEWORK

Instead of strictly enforcing DAG constraints throughout the entire iteration process, we propose a novel, scalable optimization framework that consists of three main steps:

1. Running an optimization algorithm $\mathcal{A}_1$ without any DAG constraints, only forcing the diagonal to be zero $(\mathrm{diag}(W_k) = 0)$, $\mathcal{A}_1 : \mathbb{R}^{d \times d} \to \mathbb{R}^{d \times d}$.

2. Finding a DAG that is close to the current iterate using a projection $\psi : \mathbb{R}^{d \times d} \to (\mathbb{D}, \Pi)$, which also returns its topological sorting $\pi$.

3. Running the optimization algorithm $\mathcal{A}_2$ while preserving the vertex order, $\mathcal{A}_2 : (\mathbb{D}; \Pi) \to \mathbb{D}$.

---

**Algorithm 1** $\psi$DAG framework

---

1: **Requires:** Initial model $\mathbf{W}_0 \in \mathbb{R}^{d \times d}$, such that $\mathrm{diag}(\mathbf{W}_0) = 0$.
2: **for** $k = 0, 1, 2 \ldots, K - 1$ **do**
3:     $\mathbf{W}_k^{(1/3)} = \mathcal{A}_1(\mathbf{W}_k)$                $\triangleright \mathbf{W}_k^{(1/3)} \in \mathbb{R}^{d \times d}$.
4:     $(\mathbf{W}_k^{(2/3)}, \pi_k) = \psi(\mathbf{W}_k^{(1/3)})$                $\triangleright \mathbf{W}_k^{(2/3)} \in \mathbb{D}$
5:     $\mathbf{W}_{k+1} = \mathcal{A}_2(\mathbf{W}_k^{(2/3)}; \pi_k)$                $\triangleright \mathbf{W}_{k+1} \in \mathbb{D} \subset \mathbb{R}^{d \times d}$.
6: **end for**
7: **Output:** $\mathbf{W}_K$.

---

## 4.1 OPTIMIZATION FOR THE FIXED VERTEX ORDERING

Let us clarify how to optimize while preserving the vertex order in step 3 of the framework. Given a DAG $\mathcal{G}$, we can construct its topological ordering, denoted as $ord(\mathcal{G})$. In this ordering, for every edge, the start vertex appears earlier in the sequence than the end vertex. In general, this ordering is not unique. In the space of DAGs with $d$ vertices $\mathbb{D}$, there are $d!$ possible topological orderings.

Once we have a topological ordering of the DAG, we can construct a larger DAG, $\hat{\mathcal{G}}$, by performing the transitive closure of $\mathcal{G}$. This new DAG $\hat{\mathcal{G}}$ contains all the edges of the original DAG, and additionally, it includes an edge between vertices $V_i$ and $V_j$ if there exists the path from $V_i$ to $V_j$ in $\mathcal{G}$. Thus, $\hat{\mathcal{G}}$ is an expanded version of $\mathcal{G}$.

Now, the question arises: is it possible to construct an even larger DAG that contains both $\mathcal{G}$ and $\hat{\mathcal{G}}$? The answer is yes! We call this graph the *Full DAG*, denoted by $\tilde{\mathcal{G}}$, which is constructed via full transitive closure[2]. In $\tilde{\mathcal{G}}$, there is an edge from vertex $V_i$ to vertex $V_j$ if $i < j$ in topological ordering

---

[2]Informally, for set of edges $E$, the transitive closure $E^+$ is the smallest set that includes edges $(a, b)$ whenever there is a path from $a$ to $b$ within $E$. Note that $E^+$ is the smallest superset of $E$ that satisfies that $(a, c) \in E^+$ whenever $(a, b) \in E^+, (b, c) \in E^+$.

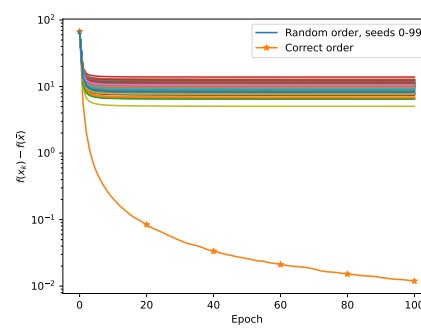 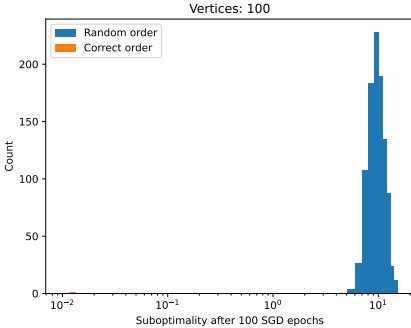

Figure 1: Minimization of (8) using SGD over a fixed topological ordering of vertices on graph type ER4 with $d = 100$ vertices with Gaussian noise. Plots demonstrate that minimizing (8) over a fixed random vertex ordering does not approach the true solution of (8).

$ord(\mathcal{G})$. This makes $\tilde{\mathcal{G}}$ the maximal DAG that includes $\mathcal{G}$. Note that for every topological sort, there is a corresponding full DAG. So, there are a total of $d!$ different full DAGs in the space of DAGs with $d$ vertices $\mathbb{D}$.

We are now ready to discuss the optimization part. Let us formulate the following optimization problem

$$\min_{\mathbf{W} \in \mathbb{R}^{d \times d}} \mathbb{E}_{x \sim \mathcal{D}} \left[ l(\mathbf{W} \cdot \mathbf{A}; x) = \tfrac{1}{2} \|x - (\mathbf{W} \cdot \mathbf{A})^\top x\|^2 \right], \qquad (9)$$

where $(\cdot)$ denotes elementwise matrix multiplication. In this formulation, $\mathbf{A}$ acts as a mask, specifying coordinates that do not require gradient computation. The problem (9) is a quadratic convex stochastic optimization problem, which can be efficiently solved using stochastic gradient descent (SGD)-type methods. These methods guarantee convergence to the global minimum, with a rate of $\mathcal{O}\left( \frac{\sigma_1 R}{\sqrt{T}} + \frac{L_1 R^2}{T} \right)$.

Assume that $\mathcal{G}^*$ is the true DAG with a weighted adjacency matrix $\mathbf{W}^*$, which is the solution we aim to find. Next, we can have the true ordering $ord(\mathcal{G}^*)$ and the true full DAG $\tilde{\mathcal{G}}^*$ with its adjacency matrix $\mathbf{A}(\tilde{\mathcal{G}}^*)$. The optimization problem (8), with the solution $\mathbf{W}^*$, can be addressed by solving the optimization problem (9) with $\mathbf{A} = \mathbf{A}(\mathcal{G}^*)$. This result indicates that, if we know the true topological ordering $ord(\mathcal{G}^*)$, then we can recover the true DAG $\mathbf{W}^*$ with high accuracy. From a discrete optimization perspective, this approach significantly reduces the space of constraints from $2^{d^2-d}$ to $d!$. To illustrate the specificity of the minimizer of the proposed problem, Figure 1 demonstrates that minimizing (8) over a fixed random vertex ordering does not approach the true solution of (8). "Correct order" curve demonstrates the convergence of (9) when the true ordering $ord(\mathcal{G}^*)$ is known.

Note that for a fixed vertex ordering and fixed adjacency matrix $\mathbf{A}$, the objective (9) becomes separable, enabling parallel computation for large-scale problems. In this work, we solved the minimization problem (9) for the number of nodes up to $d = 10^4$, at which point the limiting factor was the memory to store $\mathbf{W} \in \mathbb{R}^{d \times d}$. Through parallelization and efficient memory management, it is possible to solve even larger problems.

## 4.2 METHOD

We now introduce the method $\psi$DAG, which implements the framework outlined in Algorithm 1.

For simplicity, we select algorithm $\mathcal{A}_1$ as $\tau_1$ steps of Stochastic Gradient Descent (SGD). Similarly, $\mathcal{A}_2$ consists of $\tau_2$ steps SGD, where gradients are projected onto the space spanned by DAG's topological sorting, thus preserving the vertex order. It is important to reiterate that SGD is guaranteed to converge to the neighborhood of the solution. In the implementation, we employed an advanced version of SGD, Universal Stochastic Gradient Method from (Rodomanov et al., 2024).

The implementation of the projection method is simple as well. We compute a "closest" topological sorting and remove all edges not permitted by this ordering. The topological sorting is computed by a heuristic that calculates norms of all rows and columns to find the lowest value $v_i$. The corresponding vertex $i$ is then assigned to the ordering based on the following rule:

- If $v_i$ was the column norm, $i$ is assigned to the beginning of the ordering.

- If $v_i$ was the row norm, $i$ is assigned to the end of the ordering.

This step reduces the number of vertices, and the remaining vertices are topologically sorted using a recursive call. We formalize this procedure in Algorithm 2. Note that this procedure can be efficiently implemented without recursion and with the computation cost $\mathcal{O}(d^2)$.

---

**Algorithm 2** Projection $\psi(\mathbf{W})$ computing the "closest" vertex ordering (recursive form)

---

1: **Requires:** Model $\mathbf{W} \in \mathbb{R}^{d \times d}$, (optional) weights $\mathbf{L} \in \mathbb{R}^{d \times d}$ with default value $\mathbf{L} = \mathbf{1}\mathbf{1}^\top$.
2: **for** $k = 1, \ldots, d$ **do**
3:     Set $r_k = \| (\mathbf{W} \circ \mathbf{L}) [k][:]\|^2$
4:     Set $c_k = \| (\mathbf{W} \circ \mathbf{L}) [:][k]\|^2$
5: **end for**
6: Set $i_c = \arg\min_{k \in \{1, \ldots, d\}} c_k$
7: Set $i_r = \arg\min_{k \in \{1, \ldots, d\}} r_k$
8: **if** $r_{i_r} <= c_{i_c}$ **then**
9:     **Output:** $[\psi(\mathbf{W}(i_c, i_c), \mathbf{L}(i_c, i_c)), i_r]$
10: **else**
11:     **Output:** $[i_c, \psi(\mathbf{W}(i_c, i_c), \mathbf{L}(i_c, i_c))]$
12: **end if**
    ▷ By $A(i, j)$ we denote the submatrix $A[1, \ldots, i-1, i+1, \ldots, d][1, \ldots, j-1, j+1, \ldots, d]$

---

**Algorithm 3** $\psi$DAG

---

1: **Requires:** initial model $\mathbf{W}_0 \in \mathbb{R}^{d \times d}$, numbers or iterations $\tau_1, \tau_2$.
2: **for** $k = 0, 1, 2 \ldots, K-1$ **do**
3:     $\mathbf{W}_k^{(1/3)} = \mathsf{SGD}(\mathbf{W}_k)$                     ▷ $\tau_1$ iterations over $\mathbb{R}^{d \times d}$.
4:     $(\mathbf{W}_k^{(2/3)}, \pi_k) = $ Algorithm 2 $(\mathbf{W}_k^{(1/3)})$
5:     $\mathbf{W}_{k+1} = \mathsf{SGD}_{\pi_k}(\mathbf{W}_k)$          ▷ $\tau_2$ iterations preserving ordering $\pi_k$.
6: **end for**
7: **Output:** $\mathbf{W}_K$

---

## 5 EXPERIMENTS

We experimentally compare our newly proposed algorithm $\psi$DAG[3] to other score-based methods for computing linear DAGs, NOTEARS (Zheng et al., 2018), GOLEM[4] (Ng et al., 2020) and DAGMA (Bello et al., 2022). As it is established that DAGMA Bello et al. (2022) is an improvement over NOTEARS Zheng et al. (2018), we use mostly the former one in our experiments. As the baseline algorithms were implemented without extensive hyperparameter tuning, we avoided hyperparameter tuning as much as possible. In particular, we apply the same threshold as the one in Zheng et al. (2018), Ng et al. (2020), Bello et al. (2022) across all scenarios.

Figure 2 shows that $\psi$DAG consistently exhibits faster convergence across different noise distributions. Appendix D extends this result across different graph sizes and graph types.

---

[3]Code implementing the proposed algorithm is available at `https://anonymous.4open.science/r/psiDAG-8F42`. We use the Universal Stochastic Gradient Method from (Rodomanov et al., 2024) as the inner optimizer.

[4]In all experiments we consider GOLEM-EVwhere the noise variances are equal.

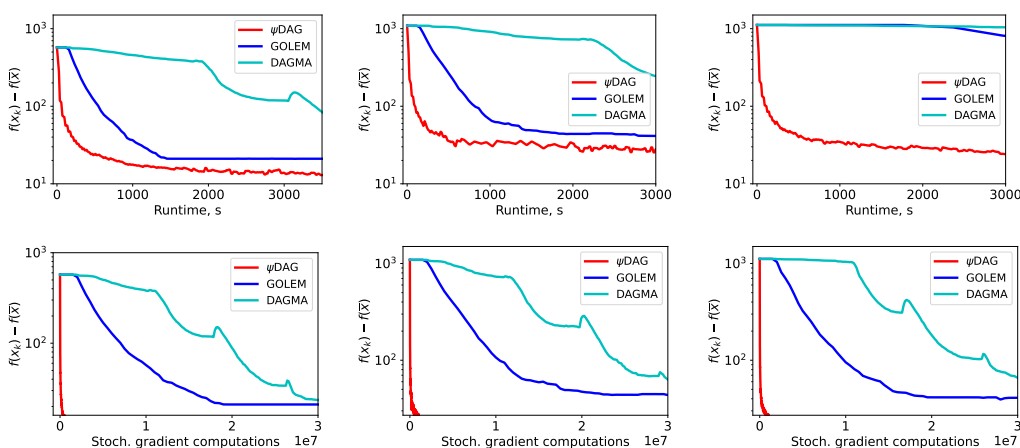

Figure 2: Linear SEM methods of $\psi$DAG, GOLEM and DAGMA on graphs of type ER4 with $d = 1000$ number of nodes and with different noise distributions: Gaussian (first), exponential (second), and Gumbel (third).

## 5.1 SYNTHETIC DATA GENERATION

We generate ground truth DAGs to have $d$ nodes and an average of $k \times d$ edges, where $k \in \{2, 4, 6\}$ is a sparsity parameter. The graph structure is determined by the choice of the graph models to be either Erdős-Rényi (ER) or Scale Free (SF), and together with sparsity parameter $k$, we refer to them as ER$k$ or SF$k$. Each of the edges has assigned a random weight uniformly sampled from the interval $[-1, -0.05] \cup [0.05, 1]$.

Following the linear Structural Equation Model (SEM), the observed data $\mathbf{X}$ has form $\mathbf{X} = \mathbf{N}(\mathbf{I} - \mathbf{W})^{-1}$, where $\mathbf{N} \in \mathbb{R}^{n \times d}$ represents $n$ $d$-dimensional independent and identically distributed (i.i.d.) noise samples drawn from either Gaussian, exponential or Gumbel distributions. In this study, we focus on an equal variance (EV) noise setting, with a scale factor of 1.0 applied to all variables. Unless otherwise specified, we generate the same number of samples $n \in \{5000, 10000\}$ for training and validation datasets, respectively. A more detailed description can be found in Appendix C.

## 5.2 SCALABILITY COMPARISON

In this section, we discuss the runtime of the proposed $\psi$DAG algorithm. We run compared algorithms until the function value converges close to the solution, $f(x_k) - f(\overline{x}) \leq 0.1 \cdot f(\overline{x})$.

Figures 3a and 3b compare the performance of $\psi$DAG against GOLEM and DAGMA on smaller graphs with various structures and noise distributions. Meanwhile, Figure 4 illustrates the scalability of $\psi$DAG on the large graphs.

Both Figures 3 and 4 clearly demonstrate that $\psi$DAG significantly outperforms GOLEM and DAGMA in terms of runtime across both sparse and dense ER graphs in nearly all considered scenarios. The only exception occurs with very small graphs $d < 100$ and high sparsity (ER2, SF2), where DAGMA is marginally faster than $\psi$DAG. However, as graph size and density increase, $\psi$DAG scale efficiently across all scenarios even up until $d = 10000$ nodes. In the case of sparse graphs (Figures 4a, 4b), $\psi$DAG consistently converges within a few hours, even for $d = 10000$ nodes.

In contrast, increasing graph size causes the computational cost of both GOLEM and DAGMA to skyrocket. Notably, across all tested graphs (4a), GOLEM exceeds allocated runtime of 36 hours for $d \geq 3000$ nodes, while DAGMA exceeds for $d \geq 5000$ nodes.

In several experimental scenarios, we also observed that both GOLEM and DAGMA occasionally failed to meet the stopping criterion, even for smaller graphs. For very small ER6 graphs ($d = 100$, Figure 4c), neither method consistently achieved the stopping criterion – DAGMA failed to converge once, and GOLEM failed twice out of three random seeds. With $d = 1000$ nodes, DAGMA again failed to converge in one out of three runs. All non-converging runs were excluded from the figures.

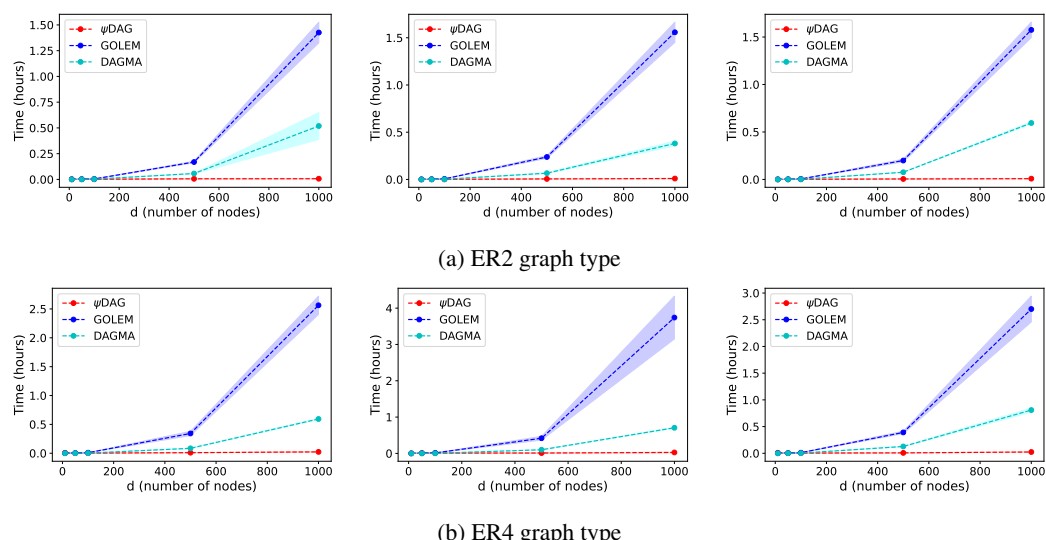

Figure 3: Runtime (hours) of $\psi$DAG, GOLEM and DAGMA for ER2 and ER4 graph types with small number of nodes $d = \{10, 50, 100, 500, 1000\}$. Noise distributions vary in different columns: Gaussian (first), exponential (second), and Gumbel (third). Method $\psi$DAG showcases much better scalability when the number of nodes increases.

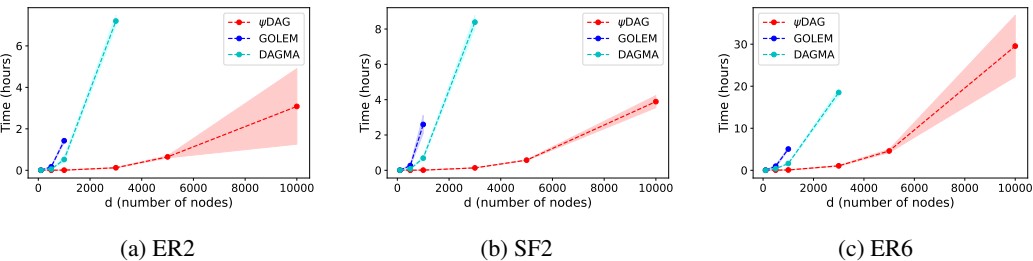

Figure 4: Runtime (hours) of $\psi$DAG, GOLEM, and DAGMA for different graph types as the graph size increases. The noise distribution is always Gaussian. Figure 4a extends Figure 3a to a large number of nodes $d \in \{3000, 5000, 10000\}$, Figure 4b presents graph type SF2 and Figure 4c showcases a more dense graph structure. Method $\psi$DAG demonstrates much better scalability as the number of nodes increases. In several scenarios, both GOLEM and DAGMA failed to consistently meet the stopping criterion. For ER6 graphs with $d = 100$ nodes, GOLEM failed to converge in two out of three runs, while DAGMA failed once. Additionally, DAGMA failed to converge in one out of three runs for $d = 1000$. All non-converging runs were excluded from the figures.

## 5.3 REAL DATA

We also evaluate the proposed method against baselines NOTEARS (Zheng et al., 2018), GOLEM (Ng et al., 2020), and DAGMA (Bello et al., 2022) on a real-world dataset, *causal protein signaling network data*, provided by Sachs et al. (2005b) that captures the expression levels of proteins and phospholipids in human cells. This dataset is widely used in the literature on probabilistic graphical models, with experimental annotations that are well-established in the biological research community.

The dataset comprises 7,466 samples, of which we utilize the first 853, corresponding to a network with 11 nodes representing proteins and 17 edges denoting their interactions. Despite its relatively small size, it is considered to be a challenging benchmark in recent studies (Zheng et al., 2018; Ng et al., 2020; Gao et al., 2021). For all experiments, we used the first 853 samples for training and the subsequent 902 samples for testing. After the training phase, we employed the same default threshold of 0.3 as was used by the other baseline approaches NOTEARS, GOLEM, DAGMA.

Table 1: Performance of the top-performing methods on the causal protein signaling network dataset Sachs et al. (2005b). The threshold for all methods is 0.3.

|  | SHD($\downarrow$) | TPR ($\uparrow$) | FPR ($\downarrow$) | Total edges | Reference |
|---|---|---|---|---|---|
| GOLEM | 26 | 0.294 | 0.47 | 23 | Ng et al. (2020) |
| NOTEARS | 15 | 0.294 | 0.26 | 15 | Zheng et al. (2018) |
| $\psi$DAG | **14** | **0.411** | **0.18** | 14 | Algorithm 3 |

As shown in Table 1, our method outperforms both baselines GOLEM (Ng et al., 2020) and NOTEARS (Zheng et al., 2018) in all metrics, the SHD (lower is better), TPR (higher is better) and FPR (smaller is better). A more detailed description can be found in Appendix C. We report the total number of edges of the output DAG. We do not report the performance of DAGMA because it fails to optimize the problem (its iterate $\mathbf{W}$ diverges from the feasible domain during the first iteration). The results for the whole dataset are shown in Appendix D.4.

## 6 CONCLUSION

We introduce a novel framework for learning Directed Acyclic Graphs (DAGs) that addresses the scalability and computational challenges of existing methods. Our approach leverages Stochastic Approximation techniques in combination with Stochastic Gradient Descent (SGD)-based methods, allowing for efficient optimization even in high-dimensional settings. A key contribution of our framework is the introduction of new projection techniques that effectively enforce DAG constraints, ensuring that the learned structure adheres to the acyclicity requirement without the need for computationally expensive penalties or constraints seen in prior works.

The proposed framework is theoretically grounded and guarantees convergence to a feasible local minimum. One of its main advantages is its low iteration complexity, making it highly suitable for large-scale structure learning problems, where traditional methods often struggle with runtime and memory limitations. By significantly reducing the per-iteration cost and improving convergence behavior, our framework demonstrates superior scalability when applied to larger datasets and more complex graph structures.

We validate the effectiveness of our method through extensive experimental evaluations across a variety of settings, including both synthetic and real-world datasets. These experiments show that our framework consistently outperforms existing methods such as GOLEM (Ng et al., 2020), NOTEARS (Zheng et al., 2018), and DAGMA (Bello et al., 2022), both in terms of objective loss and runtime, particularly in scenarios involving large and dense graphs. Furthermore, our method exhibits robust performance across different types of graph structures, highlighting its potential applicability to various practical fields such as biology, finance, and causal inference.

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

CONTENTS

## A  RELATED WORK

Zheng et al. (2018) addressed the constrained optimization problem

$$\min_{\mathbf{W} \in \mathbb{R}^{d \times d}} \ell(\mathbf{W}; \mathbf{X})_{\text{NOTEARS}} \overset{def}{=} \frac{1}{2n} \|\mathbf{X} - \mathbf{X}\mathbf{W}\|_F^2 + \lambda \|\mathbf{W}\|_1 \quad \text{subject to} \quad h(\mathbf{W}) = 0, \quad (10)$$

where $\ell(\mathbf{W}; \mathbf{X})$ represents the least squares objective and $h(\mathbf{W}) := \text{tr}(e^{\mathbf{W} \odot \mathbf{W}}) - d$ enforces the DAG constraint. Additionally, an $\ell_1$ regularization term $\lambda \|\mathbf{W}\|_1$, where $\| \cdot \|_1$ is the element-wise $\ell_1$-norm and $\lambda$ is a hyperparameter incorporated into the objective function. This formulation addresses

the linear case with equal noise variances, as discussed in Loh & Bühlmann (2014) and Peters & Bühlmann (2014). This constrained optimization problem is solved using the augmented Lagrangian method (Bertsekas et al., 1999), followed by thresholding the obtained edge weights. However, since this approach computes the acyclicity function via the matrix exponential, each iteration incurs a computational complexity of $\mathcal{O}(d^3)$, which significantly limits the scalability of the method.

Ng et al. (2020) introduced the GOLEM method, which enhances the scoring function by incorporating an additional log-determinant term, $\log|\det(\mathbf{I} - \mathbf{W})|$ to align with the Gaussian log-likelihood,

$$\min_{\mathbf{W} \in \mathbb{R}^{d \times d}} \ell(\mathbf{W}; \mathbf{X})_{\mathsf{GOLEM}} \stackrel{def}{=} \frac{d}{2} \log \|\mathbf{X} - \mathbf{X}\mathbf{W}\|_F^2 - \log|\det(\mathbf{I} - \mathbf{W})| + \lambda_1 \|\mathbf{W}\|_1 + \lambda_2 h(\mathbf{W}), \quad (11)$$

where $\lambda_1$ and $\lambda_2$ serve as regularization hyperparameters within the objective function. Although the newly added log-determinant term is zero when the current model $\mathbf{W}$ is a DAG, this score function does not provide an exact characterization of acyclicity. Specifically, the condition $\log|\det(\mathbf{I} - \mathbf{W})| = 0$ does not imply that $\mathbf{W}$ represents a DAG.

Bello et al. (2022) introduces a novel acyclicity characterization for DAGs using a log-determinant function,

$$\min_{\mathbf{W} \in \mathbb{R}^{d \times d}} \ell(\mathbf{W}; \mathbf{X})_{\mathsf{DAGMA}} \stackrel{def}{=} \frac{1}{2n} \|\mathbf{X} - \mathbf{X}\mathbf{W}\|_F^2 + \lambda_1 \|\mathbf{W}\|_1 \quad \text{subject to} \quad h_{ldet}^s(\mathbf{W}) = 0, \quad (12)$$

where $h_{ldet}^s(\mathbf{W}) \stackrel{def}{=} -\log \det(s\mathbf{I} - \mathbf{W} \circ \mathbf{W}) + d \log s$, and it is both exact and differentiable.

In practice, the augmented Lagrangian method enforces the hard DAG constraint by increasing the penalty coefficient towards infinity, which requires careful parameter fine-tuning and can lead to numerical difficulties and ill-conditioning (Birgin et al., 2005; Ng et al., 2022a). As a result, existing methods face challenges across several aspects of optimization, including the careful selection of constraints, high computational complexity, and scalability issues.

To overcome these challenges, we propose a novel framework for enforcing the acyclicity constraint, utilizing a low-cost projection method. This approach significantly reduces iteration complexity and eliminates the need for expensive hyperparameter tuning.

## B  THEORETICAL RESULTS

In this section, we present some theoretical properties of the DAG set and analyze the convergence of the proposed method.

**Lemma 2.** *The DAG set $\mathbb{D}$ is a conic set. Specifically, for any $\mathbf{W} \in \mathbb{D}$ and $\alpha \geq 0$, we have $\alpha\mathbf{W} \in \mathbb{D}$. Additionally, the DAG set $\mathbb{D}$ includes the entire line, meaning that for any $\mathbf{W} \in \mathbb{D}$ and $\alpha \in \mathbb{R}$, $\alpha\mathbf{W} \in \mathbb{D}$.*

*Proof.* We begin by observing that $\mathbf{0} \in \mathbb{D}$, as a graph with no edges is trivially a DAG. Next, consider any $\mathbf{W} \in \mathbb{D}$ and $\alpha \in \mathbb{R} \setminus \{0\}$. Scaling $\mathbf{W}$ by $\alpha$ does not alter the structure of the graph; it only changes the edge weights. Since the graph remains acyclic, $\alpha\mathbf{W} \in \mathbb{D}$. Thus, the DAG set $\mathbb{D}$ satisfies the stated properties. $\square$

Now, let us move to the subsets of DAG, which are based on a topological ordering $\pi$.

**Definition 3.** *A topological ordering $\pi$ of a directed graph is a linear ordering of its vertices such that, for every directed edge $(u, v)$ from vertex $u$ to vertex $v$, $u$ comes before $v$ in the ordering. We call $Ord(\mathbf{W})$ a set of all possible topological orderings for DAG $\mathbf{W}$ and $ord(\mathbf{W})$ is one of the orderings.*

For the graphs with $d$ vertices, there are exactly $d!$ distinct topological orderings.
Every topological ordering $\pi$ corresponds to subspace of all DAGs which can have this topological ordering, we call it $\pi$-subspace DAG.

**Definition 4.** *A $\pi$-subspace $\mathbb{D}_\pi$ is a set of all DAGs $\mathbf{W}$ such that $\pi \in Ord(\mathbf{W})$.*

Let us prove that $\pi$-subspace $\mathbb{D}_\pi$ is a linear subspace.

**Lemma 5.** $\mathbb{D}_\pi$ *is a linear subspace, meaning for any* $\mathbf{W}_1 \in \mathbb{D}_\pi, \mathbf{W}_2 \in \mathbb{D}_\pi, \alpha \in \mathbb{R}, \beta \in \mathbb{R},$ $\mathbf{W} = \alpha\mathbf{W}_1 + \beta\mathbf{W}_2 \in \mathbb{D}_\pi.$

*Proof.* We should simply note that any non-zero value in $\mathbf{W}_1$ corresponds to an edge between vertices $u$ and $v$ such that $v$ is after $u$ in the ordering $\pi$. The same holds for $\mathbf{W}_2$. Hence, any non-zero value in $\mathbf{W}$ holds the ordering $\pi$. $\square$

Next, we highlight that the DAG set $\mathbb{D}$ is a union of $\pi$-subspaces for all possible orderings $\pi$.

**Lemma 6.** *The DAG set* $\mathbb{D}$ *is a union of all* $\pi$-subspaces.

$$\mathbb{D} = \cup_\pi \mathbb{D}_\pi.$$

*Proof.* For any DAG $\mathbf{W} \in \mathbb{D}$ there exists a topological ordering $\pi$, hence $\mathbf{W} \in \mathbb{D}_\pi \in \cup_\pi \mathbb{D}_\pi$. On the other side, all elements of $\cup_\pi \mathbb{D}_\pi$ are DAGs by definition and belongs to $\mathbb{D}$.

$\square$

Now, we move to the proposed method.

**Theorem 7.** *For an* $L_1$-*smooth function* $F(\mathbf{W}) = \mathbb{E}_{x\sim\mathcal{D}}\left[l(\mathbf{W}; x)\right]$ *restricted in a domain of radius* $R$, $\|x - y\| \leq R, \forall x, y \in dom\, F$, *consider* $\mathcal{A}_2$ *in the Algorithm 1 be chosen as Universal Stochastic Gradient Method (Rodomanov et al., 2024). Running* $\mathcal{A}_2$ *for* $T$ *SGD-type steps accessing* $\sigma_1$-*stochastic gradients (Assumption 1) in the* $\pi$-*subspace* $\mathbb{D}_\pi$ *converges to a minimum of problem* (8) *with additional subspace constraints at the rate*

$$\mathbb{E}\left[F(\mathbf{W}_T) - \underset{\substack{\mathbf{W}\in\mathbb{D}_\pi, \\ ord(\mathbf{W})=\pi}}{\arg\min}\, F(\mathbf{W})\right] \leq \mathcal{O}\left(\frac{\sigma_1 R}{\sqrt{T}} + \frac{L_1 R^2}{T}\right).$$

*Proof.* A direct consequence of the convergence guarantees of the Universal Stochastic Gradient Method, Theorem 4.2 of Rodomanov et al. (2024). $\square$

**Theorem 8.** *For an* $L_1$-*smooth function* $F(\mathbf{W}) = \mathbb{E}_{x\sim\mathcal{D}}\left[l(\mathbf{W}; x)\right]$ *restricted in a domain of radius* $R$, $\|x - y\| \leq R, \forall x, y \in dom\, F$, *Algorithm Algorithm 3 with Universal Stochastic Gradient Method (Rodomanov et al., 2024) as* $\mathcal{A}_1$ *and* $\mathcal{A}_2$ *with converges to a local minimum of problem* (8).

## C  DETAILED EXPERIMENT DESCRIPTION

**Computing.** Our experiments were carried out on a machine equipped with 80 CPUs and one NVIDIA Quadro RTX A6000 48GB GPU. Each experiment was allotted a maximum wall time of 36 hours as in DAGMA Bello et al. (2022).

**Graph Models.** In our experimental simulations, we generate graphs using two established random graph models:

- **Erdős-Rényi (ER) graphs:** These graphs are constructed by independently adding edges between nodes with a uniform probability. We denote these graphs as $ER_k$, where $kd$ represents the expected number of edges.

- **Scale-Free (SF) graphs**: These graphs follow the preferential attachment process as described in Barabási & Albert (1999). We use the notation $SF_k$ to indicate a scale-free graph with expected $kd$ edges and an attachment exponent of $\beta = 1$, consistent with the preferential attachment process. Since we focus on directed graphs, this model corresponds to Price's model, a traditional framework used to model the growth of citation networks.

It is important to note that ER graphs are inherently undirected. To transform them into Directed Acyclic Graphs (DAGs), we generate a random permutation of the vertex labels from 1 to $d$, then orient the edges according to this ordering. For SF graphs, edges are directed as new nodes are added, ensuring that the resulting graph is a DAG. After generating the ground-truth DAG, we simulate the structural equation model (SEM) for linear cases, conducting experiments accordingly.

**Metrics.** The performance of each algorithm is assessed using the following four key metrics:

- **Structural Hamming Distance (SHD):** A widely used metric in structure learning that quantifies the number of edge modifications (additions, deletions, and reversals) required to transform the estimated graph into the true graph.

- **True Positive Rate (TPR):** This metric calculates the proportion of correctly identified edges relative to the total number of edges in the ground-truth DAG.

- **False Positive Rate (FPR):** This measures the proportion of incorrectly identified edges relative to the total number of absent edges in the ground-truth DAG.

- **Runtime:** The time taken by each algorithm to complete its execution provides a direct measure of the algorithm's computational efficiency.

- **Stochastic gradient computations:** Number of gradient computed.

**Linear SEM.** In the linear case, the functions are directly parameterized by the weighted adjacency matrix $W$. Specifically, the system of equations is given by $X_i = \mathbf{X}\mathbf{W}_i + N_i$, where $\mathbf{W} = [\mathbf{W}_1 | \cdots | \mathbf{W}_d] \in \mathbb{R}^{d \times d}$, and $N_i \in \mathbb{R}$ represents the noise. The matrix $\mathbf{W}$ encodes the graphical structure, meaning there is an edge $X_j \to X_i$ if and only if $W_{j,i} \neq 0$. Starting with a ground-truth DAG $B \in \{0,1\}^{d \times d}$ obtained from one of the two graph models, either ER or SF, edge weights were sampled independently from $\text{Unif}[-1, -0.05] \cup [0.05, 1]$ to produce a weight matrix $\mathbf{W} \in \mathbb{R}^{d \times d}$. Using this matrix $\mathbf{W}$, the data $X = X\mathbf{W} + N$ was sampled under the following three noise models:

- **Gaussian noise:** $N_i \sim N(0,1)$ for all $i \in [d]$,

- **Exponential noise:** $N_i \sim \text{Exp}(1)$ for all $i \in [d]$,

- **Gumbel noise:** $N_i \sim \text{Gumbel}(0,1)$ for all $i \in [d]$.

Using these noise models, random datasets $X \in \mathbb{R}^{n \times d}$ were generated by independently sampling the rows according to one of the models described above. Unless specified otherwise, each simulation generated $n = 5000$ training samples and a validation set of $10,000$ samples.

The implementation details of the baseline methods are as follows:

- NOTEARS **Zheng et al. (2018)** was implemented using the authors' publicly available Python code, which can be found at `https://github.com/xunzheng/notears`. This method employs a least squares score function, and we used their default set of hyperparameters without modification. We used the default choice of $\lambda = 0.1$ as in authors' code.

- GOLEM **Ng et al. (2020)** was implemented using the authors' Python code, available at `https://github.com/ignavierng/golem`, along with their PyTorch version at `https://github.com/huawei-noah/trustworthyAI/blob/master/gcastle/castle/algorithms/gradient/notears/torch/golem_utils/golem_model.py`. We adopted the default hyperparameter settings provided by the authors, specifically $\lambda_1 = 0.02$ and $\lambda_2 = 5$. For additional details of their method, we refer to Appendix F in Ng et al. (2020)

- DAGMA **Bello et al. (2022)** was implemented using the authors' Python code, which is available at `https://github.com/kevinsbello/dagma`. We used the default hyperparameters provided in their implementation.

**Thresholding.** Following the approach taken in previous studies, including the baseline methods (Zheng et al., 2018; Ng et al., 2020; Bello et al., 2022), for all the methods, we apply a final thresholding step of 0.3 to effectively reduce the number of false discoveries.

# D  EXPERIMENTS

Plots show performance of the algorithms $\psi$DAG, GOLEM, DAGMA for combinations of number of vertices $d \in \{10, 50, 100, 500, 1000, 3000, 5000, 10000\}$, graph types $\in \{ER, SF\}$, average density of graphs $k \in \{2, 4, 6\}$, and noise types to be either Gaussian, Exponential, or Gumbel.

Plots are grouped by the noise type and the number of vertices of the graph and arranged into figures.

**ER graph types:** Figures 5 and 6 show performance on ER2 graphs, Figures 7 and 8 show performance on ER4 graphs, Figure 9 shows performance on ER6 graphs.

**SF graph types:** Figures 10 and 11 show performance on SF2 graphs, Figures 12 and 13 show performance on SF4 graphs, Figure 14 shows performance on SF6 graphs.

We report a functional value decrease compared to **i)** time elapsed and **ii)** number of gradients computed, which also serves as a proxy of time.

Figure 6b shows that DAGMA requires a significantly larger amount of gradient computations compared to both $\psi$DAG and GOLEM.

## D.1  SMALL TO MODERATE NUMBER OF NODES

Our experiments demonstrate that while number of nodes is small, $d < 100$, GOLEM is more stable than DAGMA, and $\psi$DAG method is the most stable. While DAGMA shows impressive speed for smaller node sets, the number of iterations required is still higher than both GOLEM and our method. Across all scenarios, $\psi$DAG consistently demonstrates faster convergence compared to the other approaches, requiring fewer iterations to reach the desired solution.

## D.2  LARGE NUMBER OF NODES

For graphs with a large number of nodes $d \in \{5000, 10000\}$, we were unable to run neither of the baselines, and consequently, Figure 15 includes only one algorithm. GOLEM was not feasible due to its computation time exceeding 350 hours. DAGMA was impossible as its runs led to kernel crashes. In all cases, we utilized a training set of 5,000 samples and a validation set of 10,000 samples.

## D.3  DENSER GRAPHS

For a thorough comparison, in Figures 9 and 14, we compare graph structures ER6 and SF6 under the Gaussian noise type. Plots indicate that while DAGMA exhibits a fast runtime when the number of nodes is small, $d < 100$, it requires more iterations to achieve convergence. Algorithm $\psi$DAG consistently outperforms GOLEM and DAGMA in both training time and a number of stochastic gradient computations, and the difference is more pronounced for a larger number of nodes and denser graphs.

## D.4  REAL DATA

In addition to demonstrating the effectiveness of our method on the most challenging subset of the dataset Sachs et al. (2005b), we also evaluated its performance using the entire dataset of $n = 7,466$ samples for training. As shown in Table 2, our method consistently outperforms the baselines GOLEM and NOTEARS in terms of the FPR metric while achieving the same SHD as NOTEARS and matching the TPR across all methods. Moreover, similar to previous results, DAGMA fails to optimize the problem when applied to the entire dataset, further highlighting the robustness and reliability of our approach.

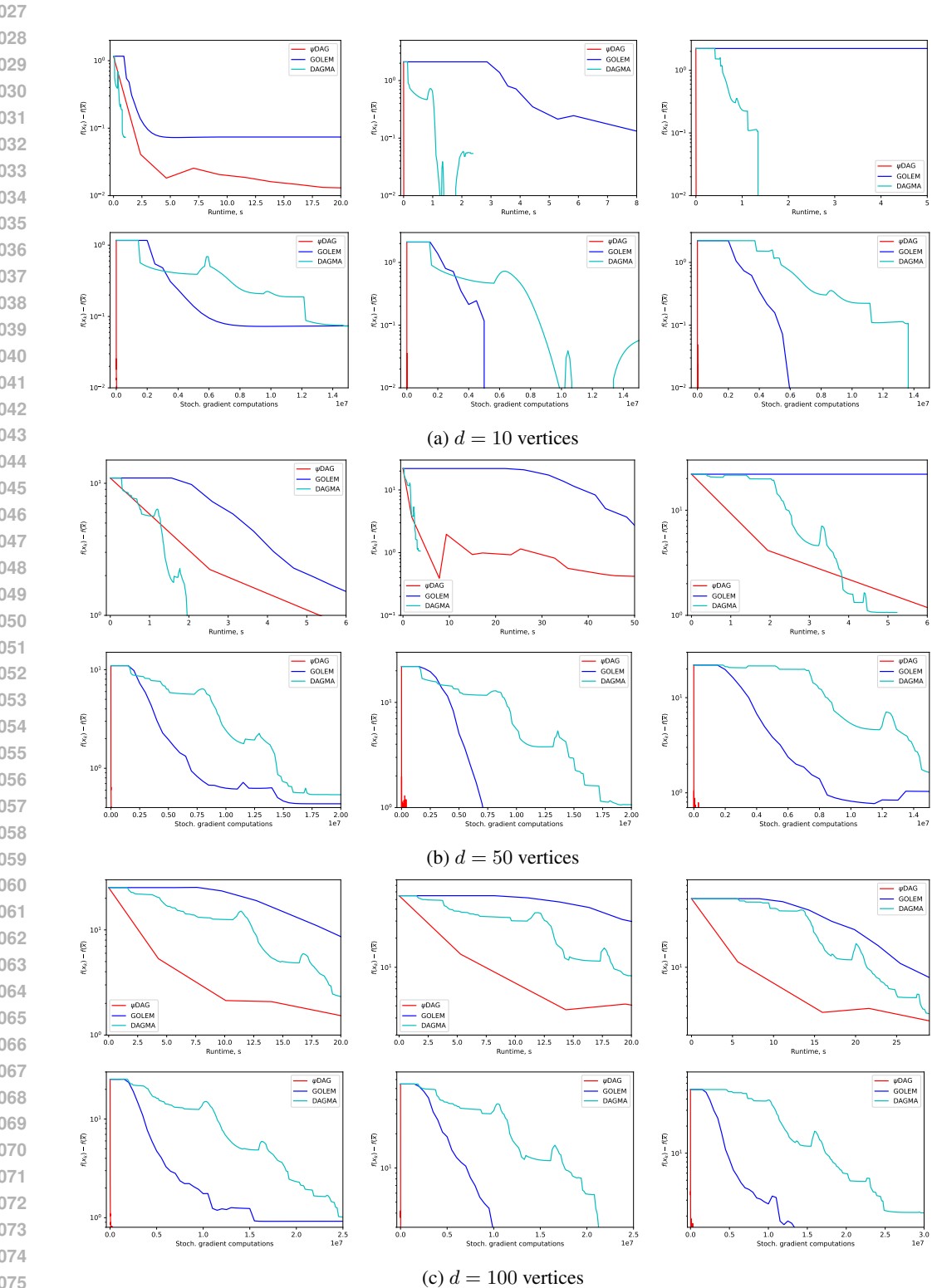

(a) $d = 10$ vertices

(b) $d = 50$ vertices

(c) $d = 100$ vertices

Figure 5: Linear SEM methods on graphs of type ER2 with different noise distributions: Gaussian (first), exponential (second), Gumbel (third).

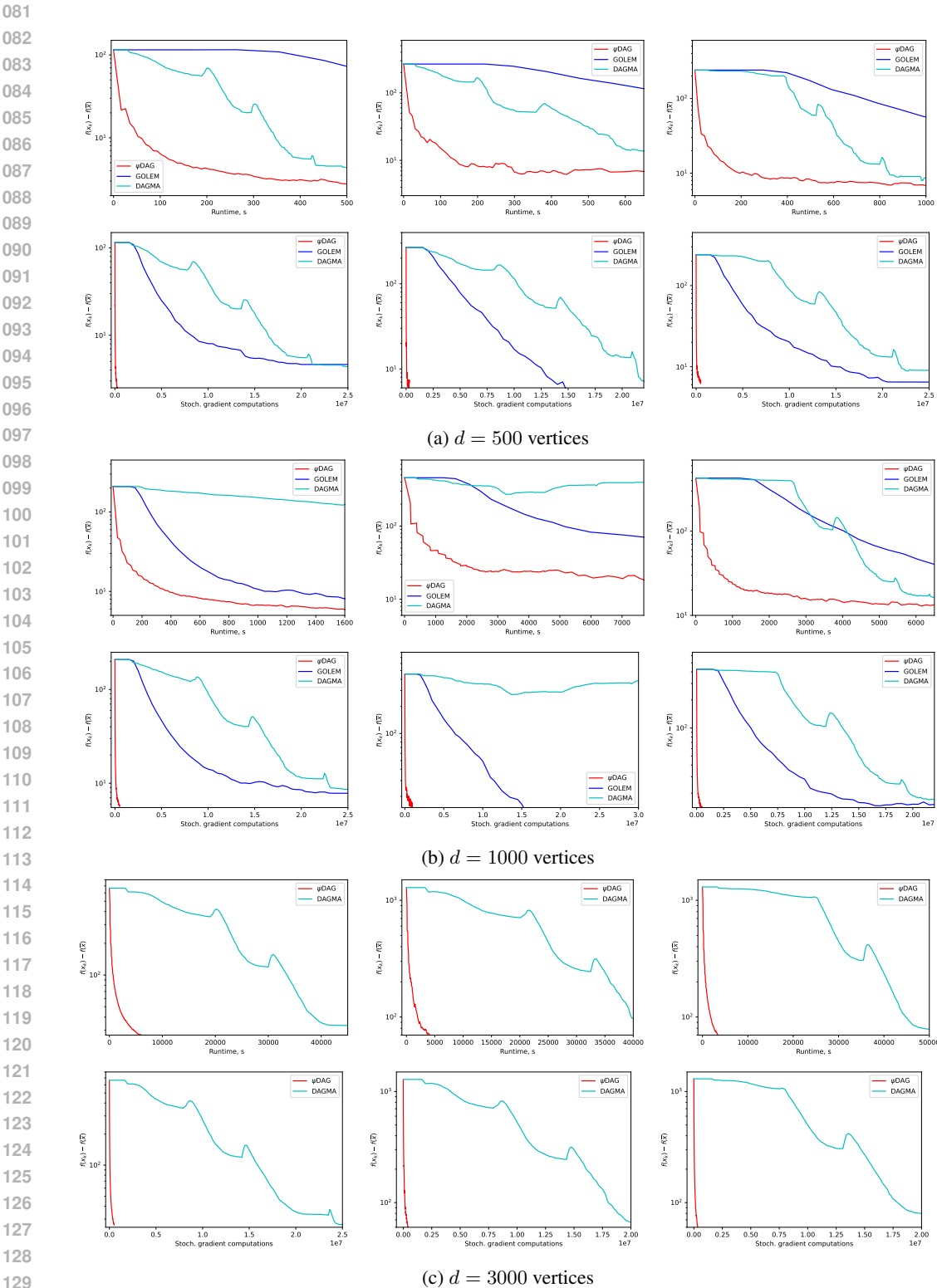

(a) $d = 500$ vertices

(b) $d = 1000$ vertices

(c) $d = 3000$ vertices

Figure 6: Linear SEM methods on graphs of type ER2 with different noise distributions: Gaussian (first), exponential (second), Gumbel (third).

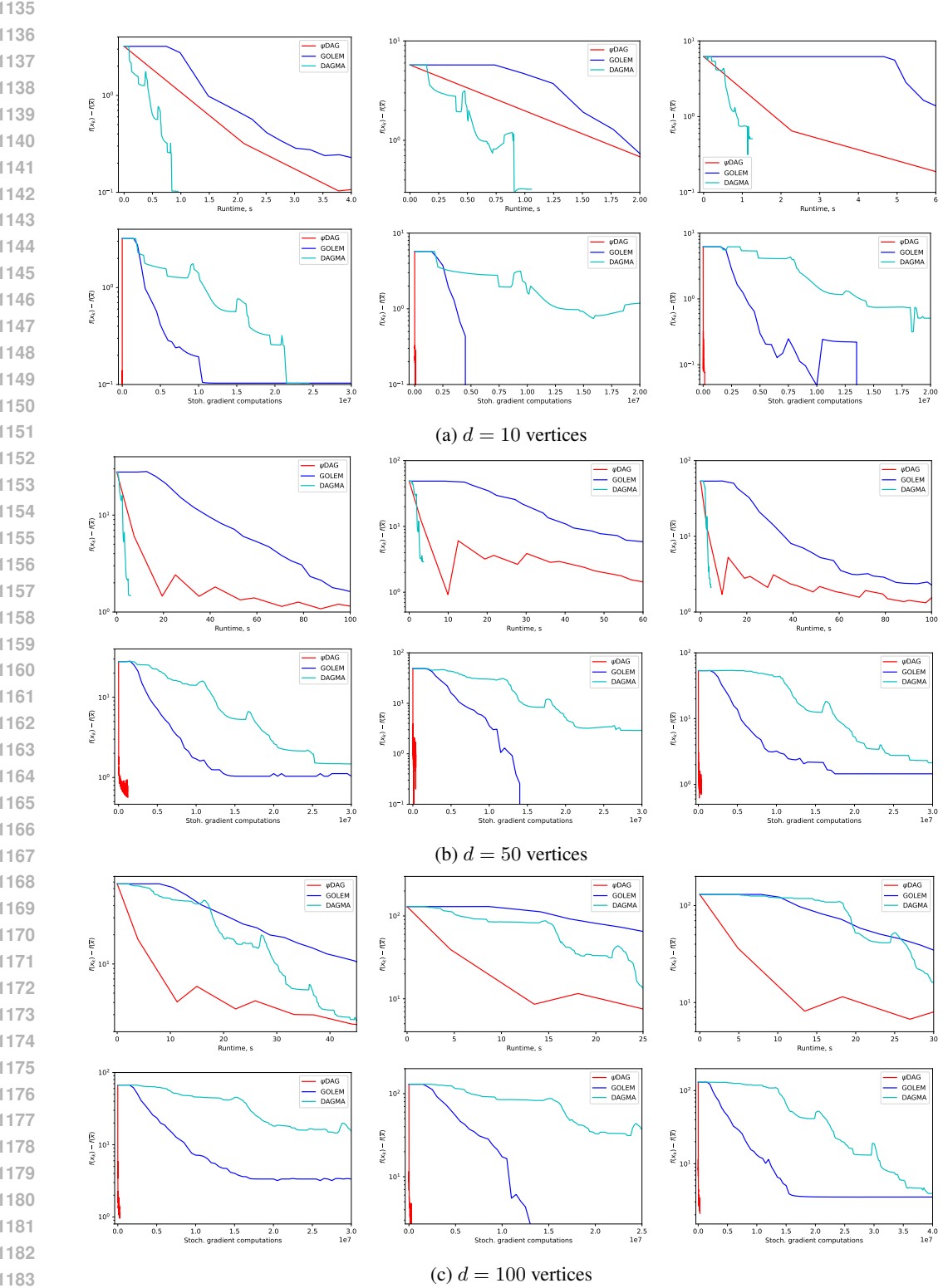

(a) $d = 10$ vertices

(b) $d = 50$ vertices

(c) $d = 100$ vertices

Figure 7: Linear SEM methods on graphs of type ER4 with different noise distributions: Gaussian (first), exponential (second), Gumbel (third).

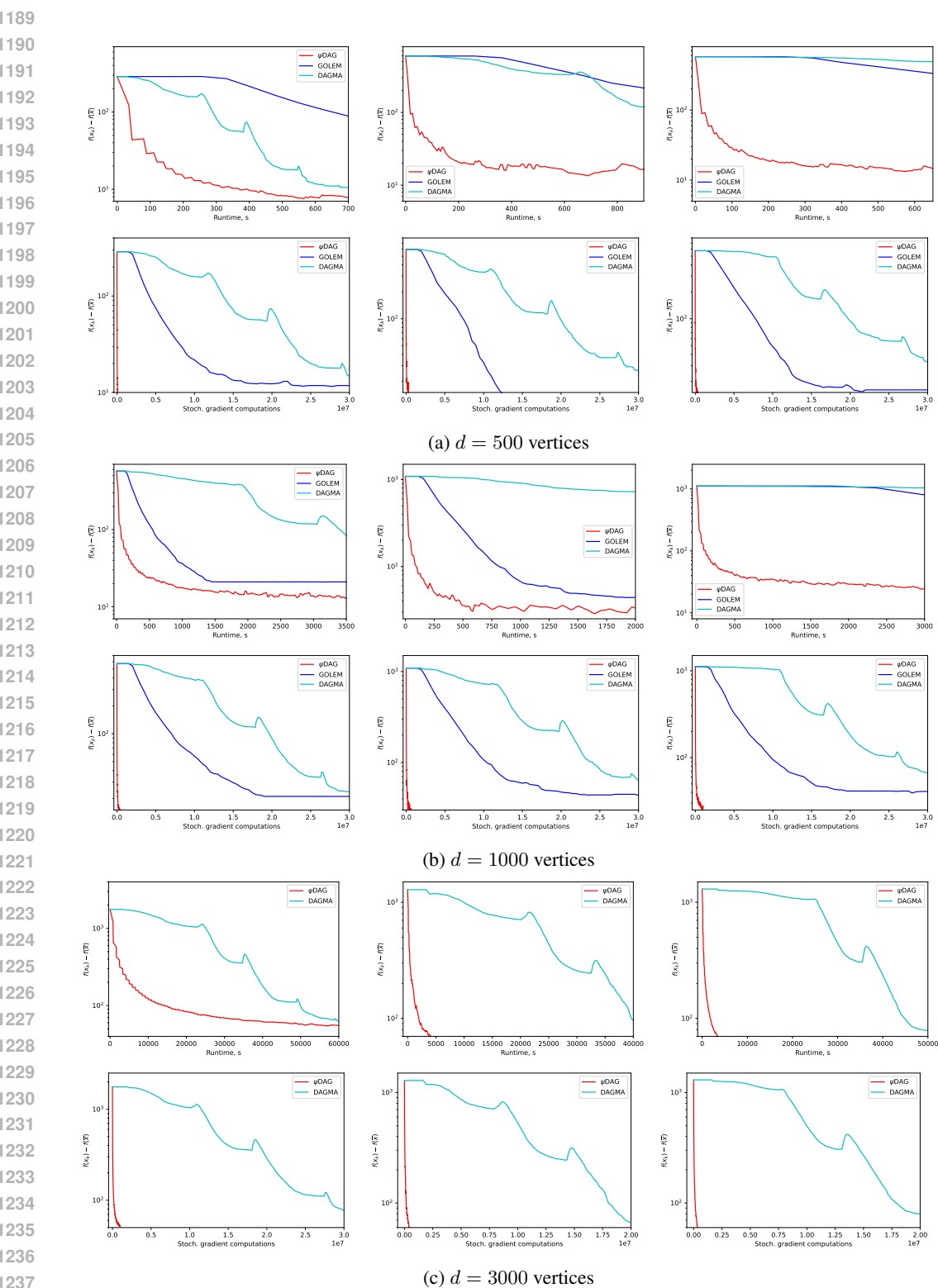

(a) $d = 500$ vertices

(b) $d = 1000$ vertices

(c) $d = 3000$ vertices

Figure 8: Linear SEM methods on graphs of type ER4 with different noise distributions: Gaussian (first), exponential (second), Gumbel (third).

Table 2: Performance of the top-performing methods on the causal protein signaling network dataset Sachs et al. (2005b). The threshold for all methods is 0.3.

|  | SHD($\downarrow$) | TPR ($\uparrow$) | FPR ($\downarrow$) | Total edges | Reference |
|---|---|---|---|---|---|
| GOLEM | 21 | **0.29** | 0.39 | 20 | Ng et al. (2020) |
| NOTEARS | **19** | **0.29** | 0.39 | 21 | Zheng et al. (2018) |
| $\psi$DAG | **19** | **0.29** | **0.34** | 17 | Algorithm 3 |

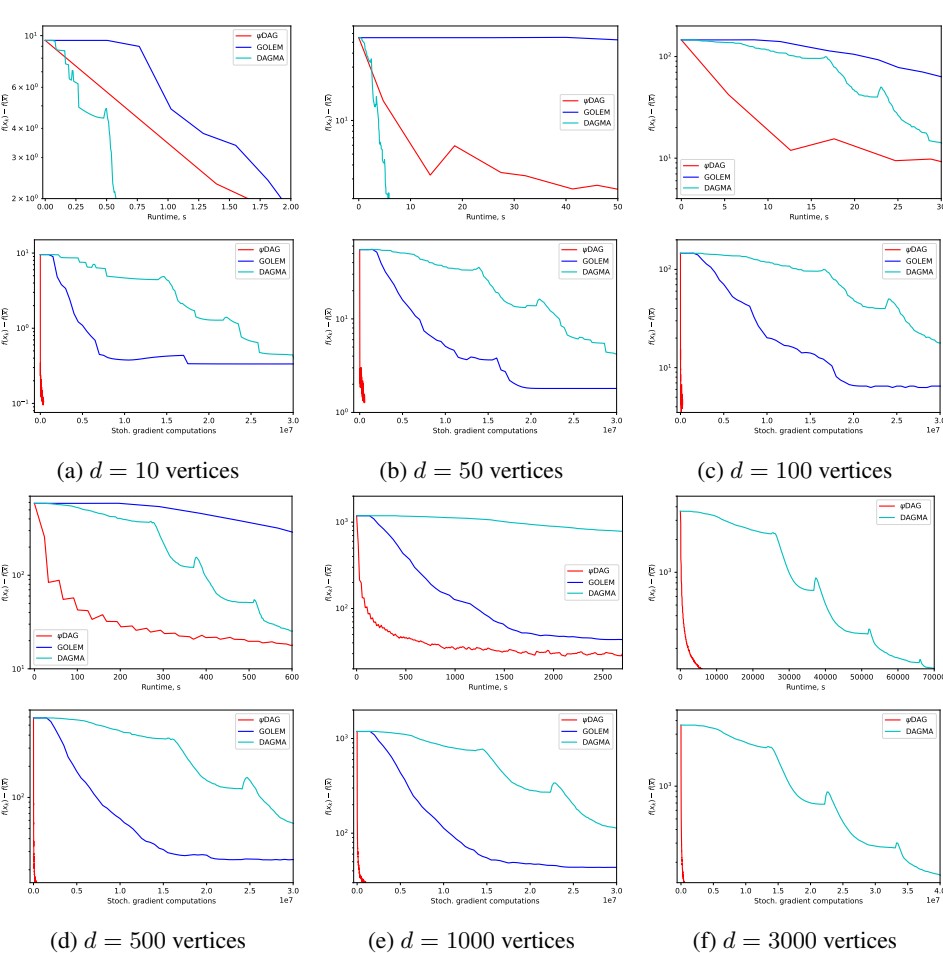

(a) $d = 10$ vertices     (b) $d = 50$ vertices     (c) $d = 100$ vertices

(d) $d = 500$ vertices     (e) $d = 1000$ vertices     (f) $d = 3000$ vertices

Figure 9: Linear SEM methods on graphs of type ER6 with the Gaussian noise distribution.

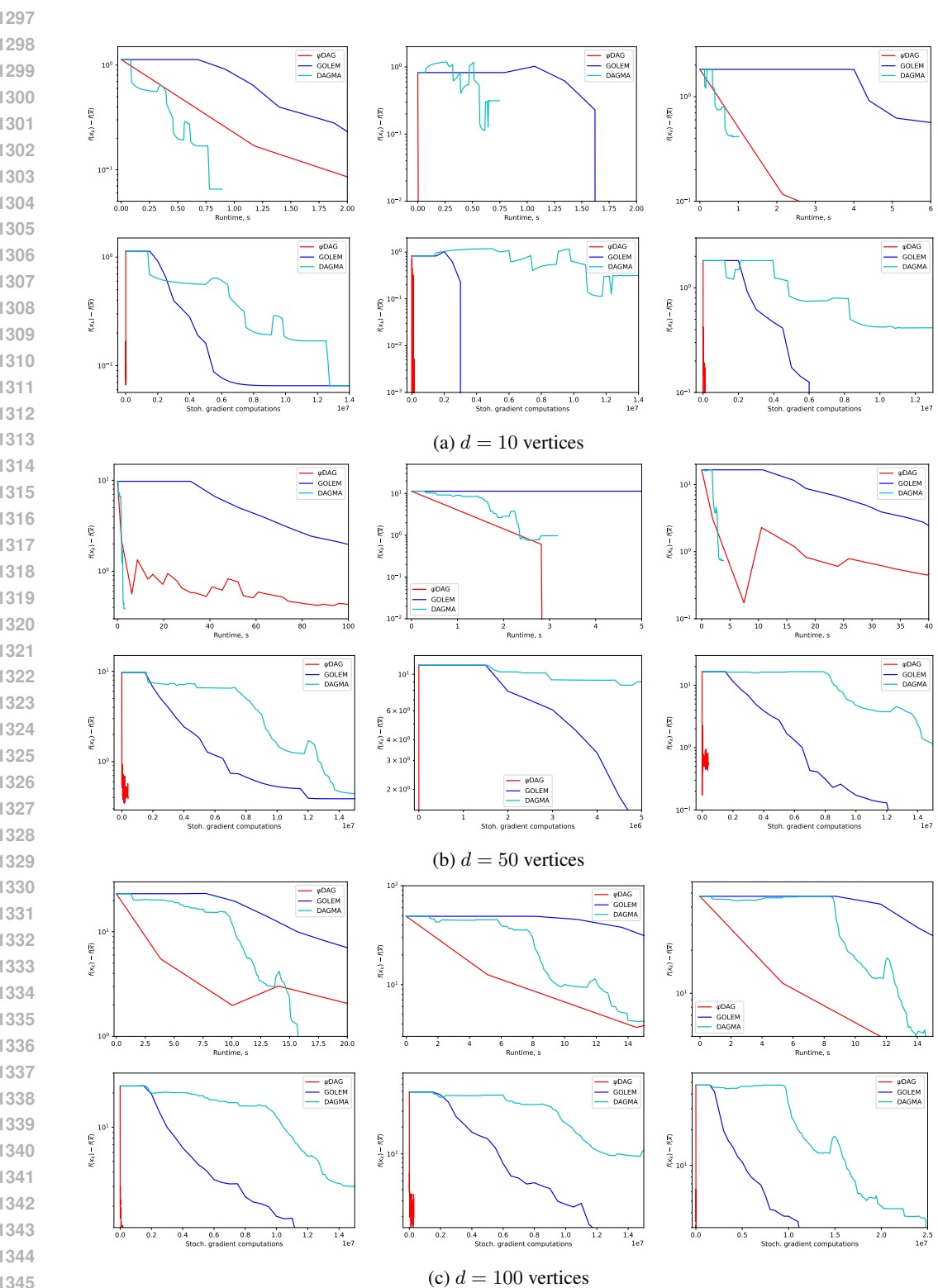

(a) $d = 10$ vertices

(b) $d = 50$ vertices

(c) $d = 100$ vertices

Figure 10: Linear SEM methods on graphs of type SF2 with different noise distributions: Gaussian (first), exponential (second), Gumbel (third).

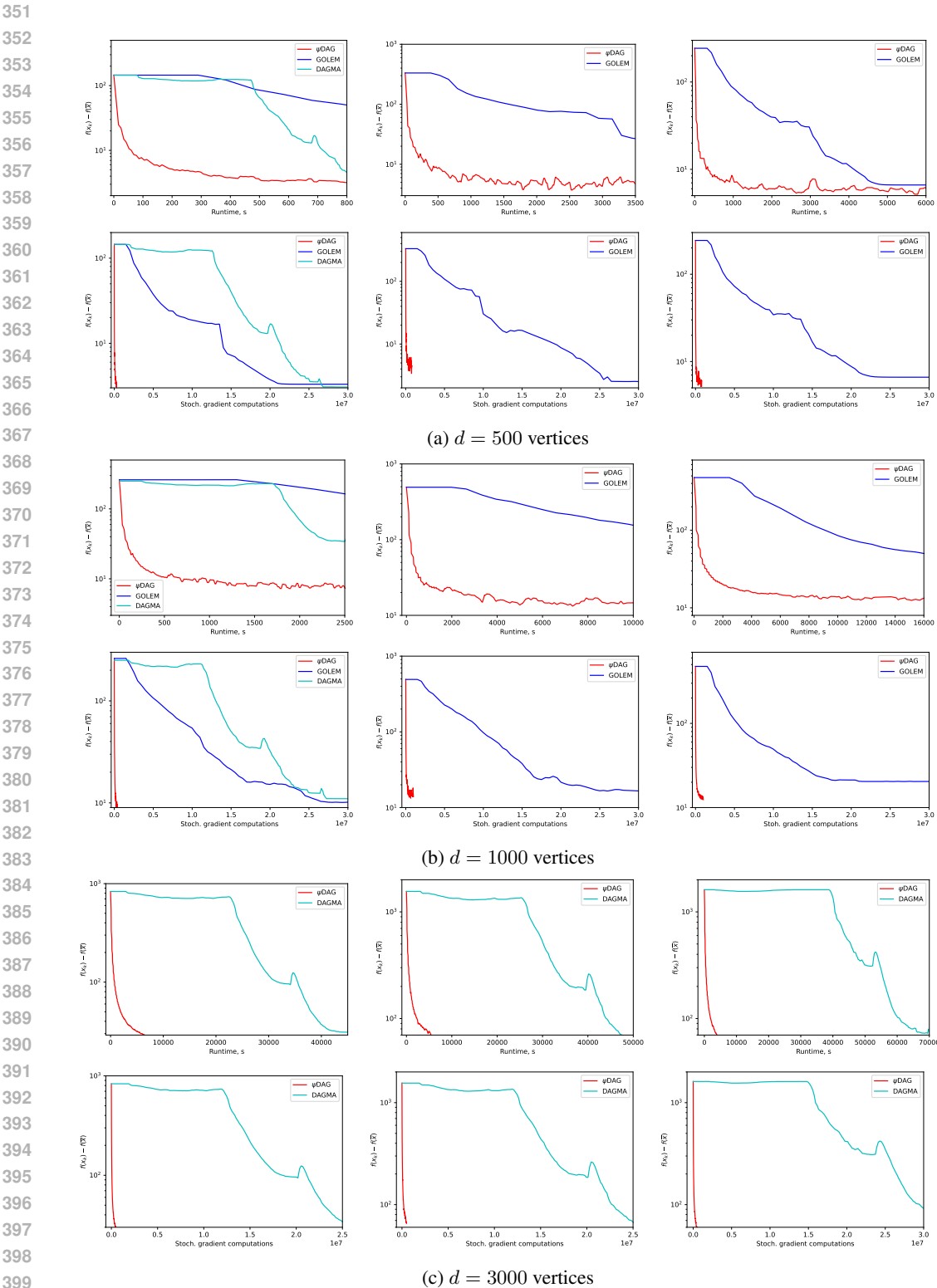

(a) $d = 500$ vertices

(b) $d = 1000$ vertices

(c) $d = 3000$ vertices

Figure 11: Linear SEM methods on graphs of type SF2 with different noise distributions: Gaussian (first), exponential (second), Gumbel (third).

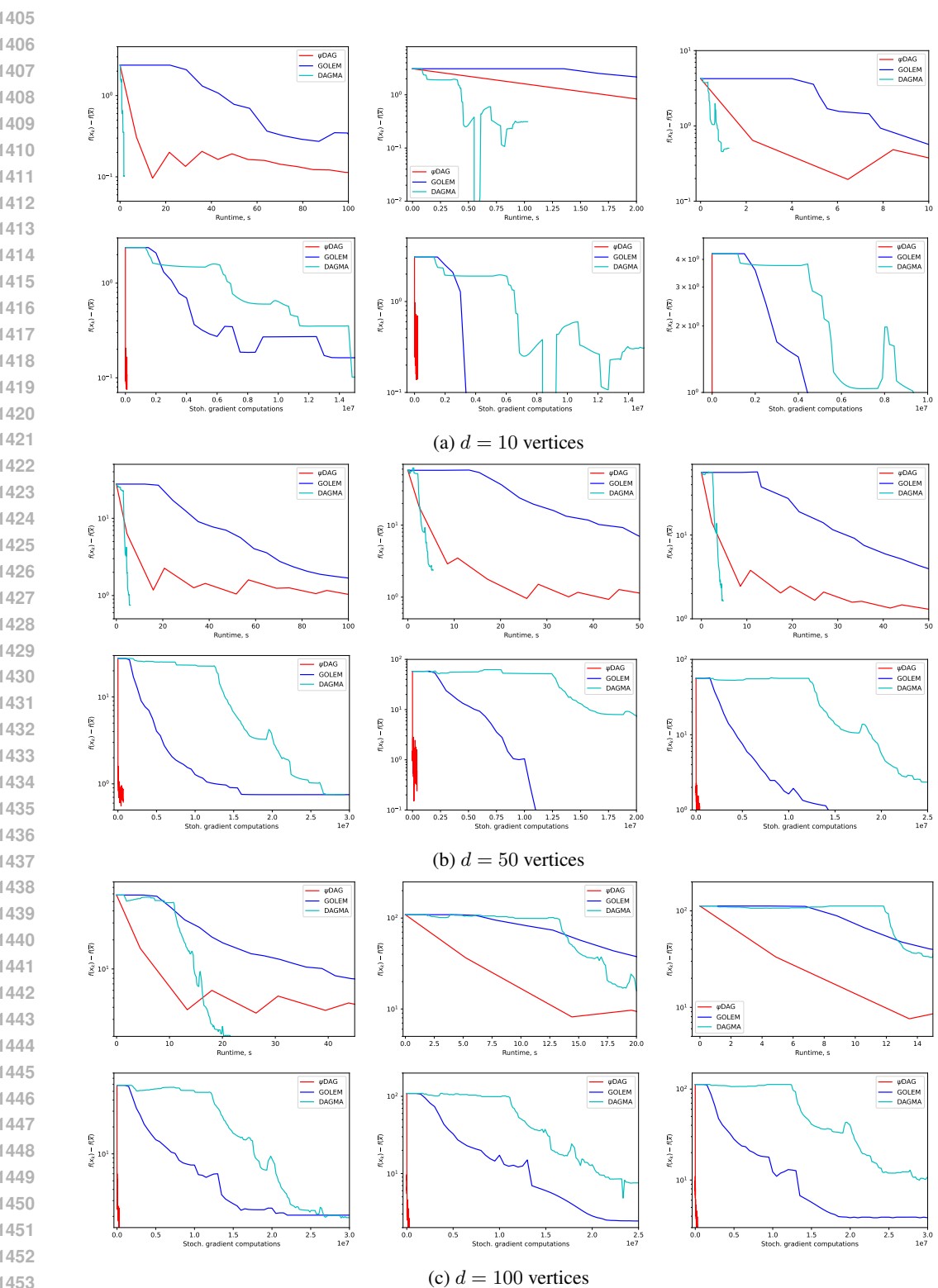

(a) $d = 10$ vertices

(b) $d = 50$ vertices

(c) $d = 100$ vertices

Figure 12: Linear SEM methods on graphs of type SF4 with different noise distributions: Gaussian (first), exponential (second), Gumbel (third).

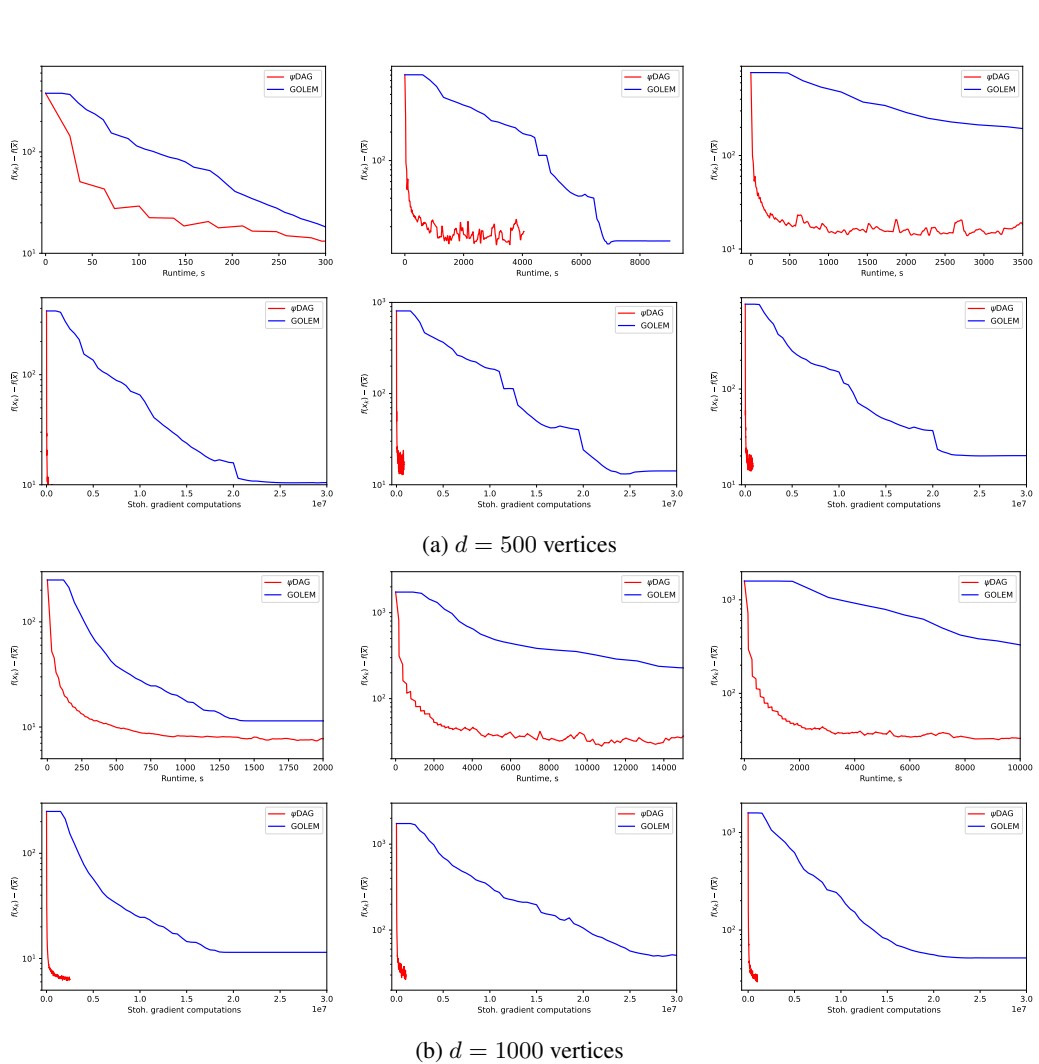

(a) $d = 500$ vertices

(b) $d = 1000$ vertices

Figure 13: Linear SEM methods on graphs of type SF4 with different noise distributions: Gaussian (first), exponential (second), Gumbel (third).

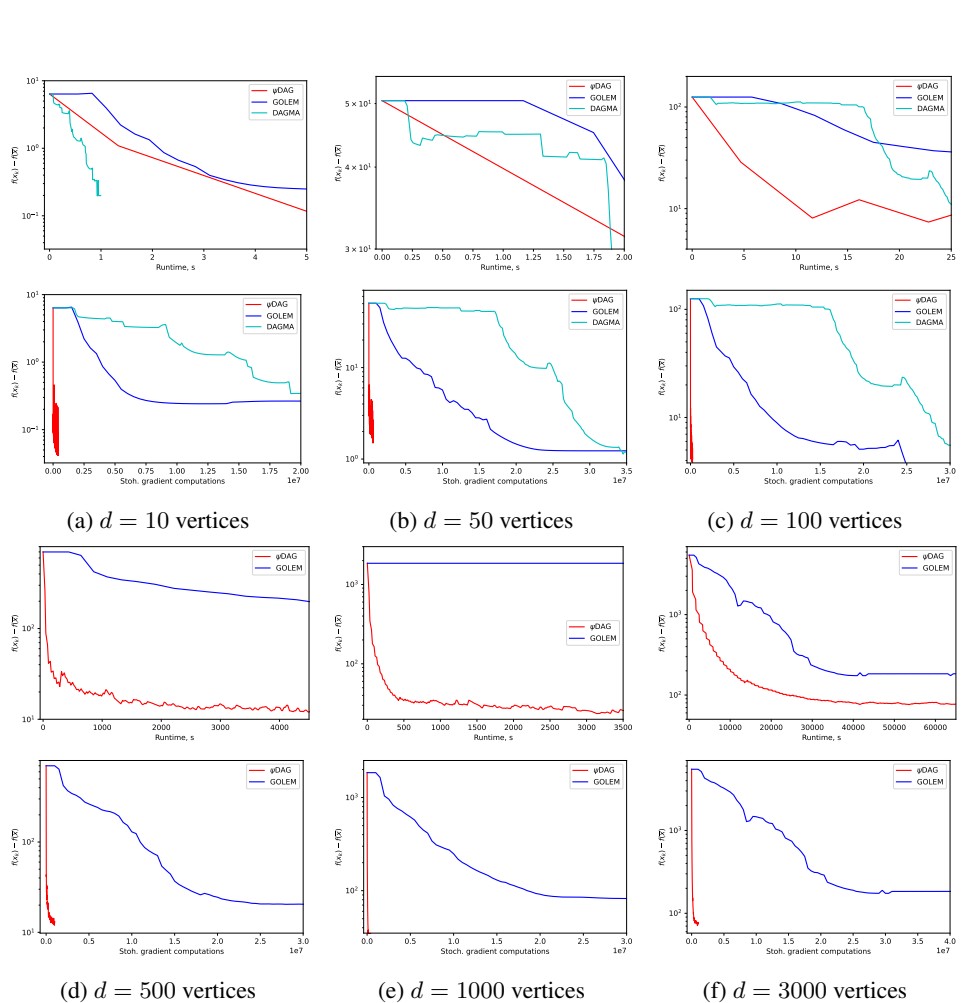

(a) $d = 10$ vertices  (b) $d = 50$ vertices  (c) $d = 100$ vertices

(d) $d = 500$ vertices  (e) $d = 1000$ vertices  (f) $d = 3000$ vertices

Figure 14: Linear SEM methods on graphs of type SF6 with the Gaussian noise distribution.

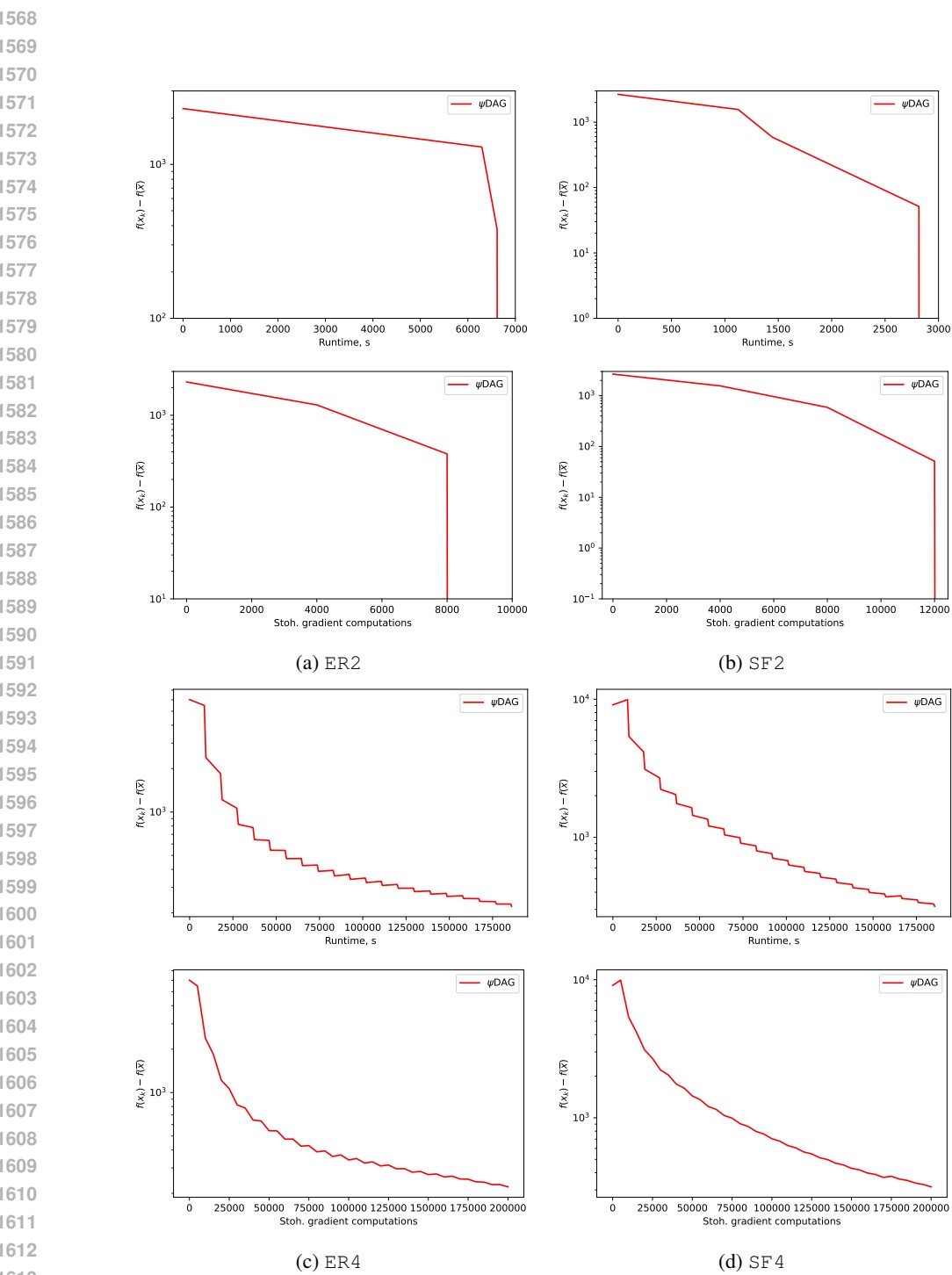

Figure 15: $\psi$DAG method for graph types ER2, ER4, SF2 and SF4 graphs with $d = 10000$ and Gaussian noise. Other linear SEM methods do not converge in less than $350$ hours.

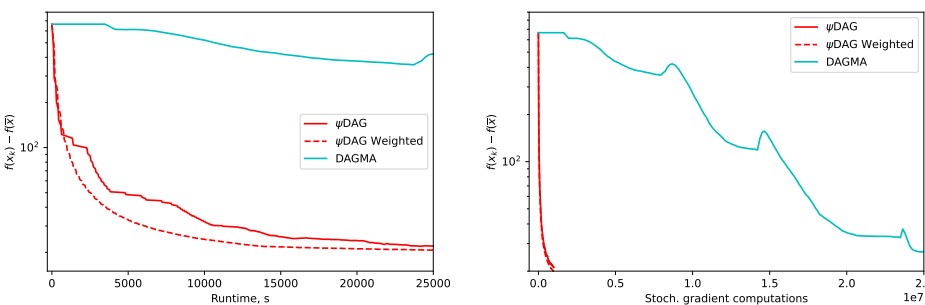

Figure 16: Comparison of $\psi$DAG, $\psi$DAG weighted and DAGMA for ER2 graph with $d = 3000$ nodes and Gaussian noise.

## E  WEIGHTED PROJECTION

Inspired by the importance sampling, we considered adjustment of the projection method by weights. Specifically, we considered the elements of the $\mathbf{W}$ to be weighted element-wisely by the second directional derivatives of the objective function, $\mathbf{L}[i][j] \stackrel{def}{=} \left(\frac{d}{d\mathbf{W}[i][j]}\right)^2 \mathbb{E}_{X \sim \mathcal{D}}\left[l(\mathbf{W}; X)\right]$. As we don't have access to the whole distribution $\mathcal{D}$, we approximate it by the mean of already seen samples,

$$\mathbf{L}_k[i][j] \stackrel{def}{=} \left(\frac{d}{d\mathbf{W}[i][j]}\right)^2 \frac{1}{k} \sum_{k=0}^{k-1} l\left(\mathbf{W}; X_k\right) = \frac{1}{k} \sum_{t=0}^{k-1} \left(X_k[j]\right)^2. \tag{13}$$

Weights (13) are identical for whole columns; hence, they impose storing only one vector. Updating them requires a few element-wise vector operations.

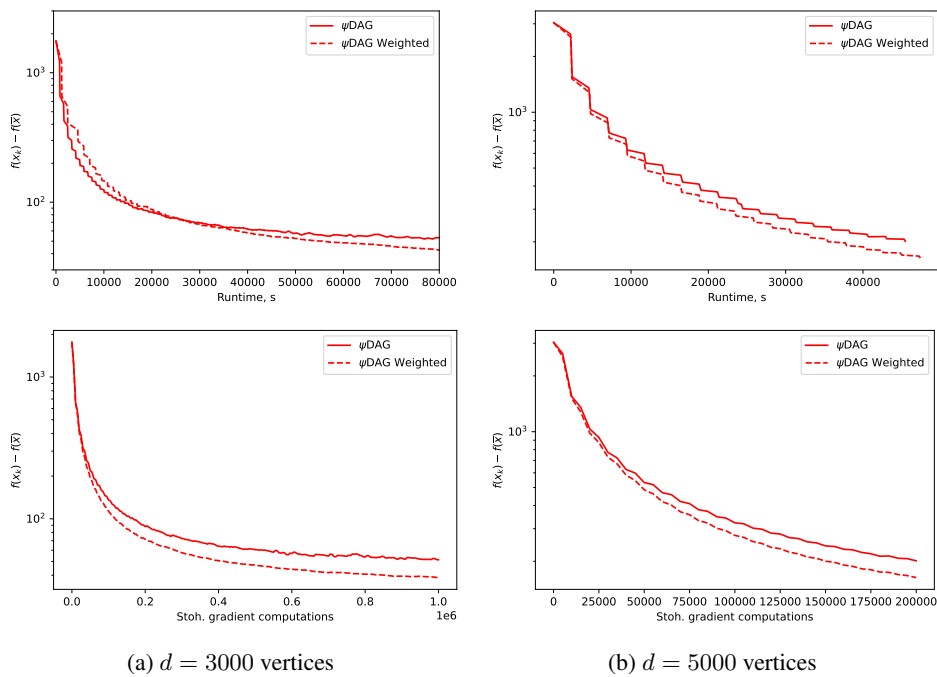

(a) $d = 3000$ vertices $\qquad\qquad$ (b) $d = 5000$ vertices

Figure 17: $\psi$DAG method with weighted projection for graph types ER4 and Gaussian noise.

Figures 16 and 17 show that this weighting can lead to an improved convergence (slightly faster convergence to a slightly lower functional value) without imposing any extra gradient computation. However, we noticed that the improvement over runtime is not consistent across different experiments; hence, for simplicity, we deferred this to the appendix.