# OpenReview forum: "$\psi$DAG : Projected Stochastic Approximation Iteration for DAG Structure Learning"
_ICLR.cc/2025/Conference — Submitted to ICLR 2025_

### Official Review · Reviewer_pNVL · 2024-10-27

**Soundness:** 2
**Presentation:** 2
**Contribution:** 2
**Rating:** 6
**Confidence:** 4

**Summary:**

The paper leverages stochastic optimization within a newly proposed framework for DAG learning, introducing a novel algorithm. The authors effectively demonstrate the efficiency of this algorithm through comprehensive experimentation.

**Strengths:**

This paper is the first I have encountered in differentiable DAG learning that integrates a stochastic algorithm, presenting a novel and compelling approach. A primary advantage of the stochastic algorithm lies in its computational efficiency, offering a marked improvement over conventional methods. The authors have also conducted an extensive set of experiments to substantiate the algorithm's efficiency, which I find to be thorough and convincing.

**Weaknesses:**

The paper’s approach is limited by its reliance on a simple linear model with equal variance noise, resulting in an easy optimization problem (least squares with $\ell_1$ penalty). When the graph order is fixed, the problem reduces to a Lasso optimization, which is computationally trivial for large dimensions (e.g., $d =1000,n =5000$), taking only a few seconds or minutes to solve. This simplicity undermines the significance of the study. To make the work more meaningful, the authors might consider applying their stochastic algorithm to more complex structural equation models (SEM), where optimization challenges are more substantial.


Additionally, several statements in the paper lack rigor. For example, the abstract and Theorem 2 claim that the algorithm converges to a feasible local minimum, but no theoretical proof or justification is provided for Algorithm 1, which is crucial given the NP-complete nature of the problem in (11). Original guarantee of local optimal solution from [1][2] can not be directly applied here, since it is unconstrained optimization and objective function is convex. Please refer to [3][4], these two paper study the local optimality of problem (11) indetails, they should be included and discussed appropirately in the paper. Furthermore, the transitive closure of $\mathcal{G}$ in lines 261-262 lacks a formal definition, which should be added for clarity.

The experimental setup also introduces concerns. The authors sample edge weights from a non-standard range, uniformly from $[−1, −0.05] \cup [0.05, 1]$, which deviates from established practices. Applying the same threshold (0.3) as methods like NOTEARS, DAGMA, and GOLEM in this unconventional setting risks removing correctly recovered edges, potentially skewing baseline performance and inflating the loss gap. To address this, the authors should consider aligning with the edge weight sampling practices of prior studies to ensure comparability. Additionally, incorporating Structural Hamming Distance (SHD) as a recovery accuracy measure and including error bars to capture result variability would improve the robustness and interpretability of findings.

[1] Arkadi Nemirovski and David Yudin, Problem Complexity and Method Efficiency in Optimization
[2] Guanghui Lan, An optimal method for stochastic composite optimization
[3] Dennis Wei, Tian Gao, Yue Yu, DAGs with No Fears: A Closer Look at Continuous Optimization for Learning Bayesian Networks
[4] C Deng, K Bello, B Aragam, PK Ravikumar, Optimizing NOTEARS objectives via topological swaps
[4] PL Loh, P Bühlmann, High-dimensional learning of linear causal networks via inverse covariance estimation

**Questions:**

No further quesions, questions are addressed above.

---

> ### Author Response · Authors · 2024-11-19
> **Rebuttal part I**
>
> We sincerely thank the reviewer for their helpful comments and would like to provide answers to all the concerns raised.
>
> ----
> > _The paper’s approach is limited by its reliance on a simple linear model with equal variance noise, resulting in an easy optimization problem (least squares with  penalty). When the graph order is fixed, the problem reduces to a Lasso optimization, which is computationally trivial for large dimensions (e.g., d =1000, d =5000), taking only a few seconds or minutes to solve. This simplicity undermines the significance of the study. To make the work more meaningful, the authors might consider applying their stochastic algorithm to more complex structural equation models (SEM), where optimization challenges are more substantial._
>
> **Answer:** As we have demonstrated in Figure 1, a solution of the optimization problem over a fixed (random) graph ordering is far from the solution of the optimization problem minimized over all DAGs. Therefore, we claim that the results of our paper are already meaningful.
>
> We restricted our setup to linear SEMs just for the sake of simplicity. Extensions to nonlinear SEMs are possible and are the focus of our ongoing research.
>
> ___
> >_Additionally, several statements in the paper lack rigor. For example, the abstract and Theorem 2 claim that the algorithm converges to a feasible local minimum, but no theoretical proof or justification is provided for Algorithm 1, which is crucial given the NP-complete nature of the problem in (11)._
>
> **Answer:** We would like to apologize for the missing proof of Theorem 2. The proof is a direct consequence of the application of the algorithm Universal Stochastic Gradient Method (Alg.3 of [SGD]).
>
> In particular, Theorem 4.2 of [SGD] guarantees that applied to convex L-Hölder continuous functions in the domain of radius $R$, $||x-y|| \leq R$, in $k$ steps decreases functional loss at rate $\mathcal{O} (\frac{LR}{k^2} + \frac {\sigma R}{\sqrt k} )$.
> In our case, once our Algorithm 1 fixes the ordering $\pi_k$ in line 4, DAGs allowed by ordering $\pi_k$ form a linear subspace of $R^{d \times d}$ and in that subspace loss (11) is smooth and convex, and $k$ steps Universal Stochastic Gradient Method converge to a minimum of the ordering at rate given by Theorem 4.2 of [SGD]. This minimum of the ordering is a local minimum of the loss (11).
>
> [SGD] Universal Gradient Methods for Stochastic Convex Optimization, Anton Rodomanov, Ali Kavis, Yongtao Wu, Kimon Antonakopoulos, Volkan Cevher, link:https://openreview.net/pdf?id=Wnhp34K5jR
>
> ___
> >_...Please refer to [3][4], these two paper study the local optimality of problem (11) indetails, they should be included and discussed appropirately in the paper._
>
> **Answer:** Thank you. We will include these papers in our paper.
>
> Reference [3] investigates the NOTEARS framework for continuous optimization in Bayesian network structure learning, providing new theoretical insights into the KKT conditions for such formulations. They propose combining NOTEARS with a local search method informed by KKT conditions (KKTS), leading to improvements in SHD.
>
> While [3] focuses on enhancing existing algorithms with a post-processing step, our work takes an orthogonal approach, reformulating DAG learning as a stochastic optimization problem. We develop an entirely new framework that seamlessly incorporates stochastic gradient methods, allowing efficient processing of large graphs (up to 10,000 nodes), exceeding the computational limits of NOTEARS and [3].
>
> Reference [4] presents a bi-level algorithm that iteratively refines topological orders through node swaps. By combining off-the-shelf linear constraint solvers with a novel method for identifying node pairs for swaps, it finds local minima or KKT points under relaxed conditions, achieving solutions with lower scores than state-of-the-art methods. This approach can also serve as a post-processing step to refine the outputs of other algorithms.
>
> Comparing our work to [4], we again adopt an orthogonal approach by reformulating a minimization over DAGs as a stochastic optimization task. While the topological swaps method emphasizes refining solutions and improving local optimality, our work addresses the challenges of large-scale graph learning, providing a scalable framework.

---

> > ### Author Response · Authors · 2024-11-19
> > **Rebuttal part II**
> >
> > >_The experimental setup also introduces concerns. The authors sample edge weights from a non-standard range, uniformly from $[-1, -0.05] \cup [0.05, 1]$, which deviates from established practices. Applying the same threshold (0.3) as methods like NOTEARS, DAGMA, and GOLEM in this unconventional setting risks removing correctly recovered edges, potentially skewing baseline performance and inflating the loss gap. To address this, the authors should consider aligning with the edge weight sampling practices of prior studies to ensure comparability. Additionally, incorporating Structural Hamming Distance (SHD) as a recovery accuracy measure and including error bars to capture result variability would improve the robustness and interpretability of findings._
> >
> > **Answer:** We would like to point out that the choice of thresholding does not affect the optimization process and can be adjusted (or completely removed) without affecting the framework.
> >
> > While other related works often use thresholding value 0.3 for true weights in range $[-2, -0.5] \cup [0.5, 2]$, we consciously chose to deviate from this practice. Such a choice of thresholding implies knowledge of the sampling boundaries, which we consider to be unrealistic in many practical scenarios.

---

> > ### Comment · Reviewer_pNVL · 2024-11-22
> >
> > > Answer: As we have demonstrated in Figure 1, a solution of the optimization problem over a fixed (random) graph ordering is far from the solution of the optimization problem minimized over all DAGs. Therefore, we claim that the results of our paper are already meaningful.
> >
> > > We restricted our setup to linear SEMs just for the sake of simplicity. Extensions to nonlinear SEMs are possible and are the focus of our ongoing research.
> >
> > In your algorithm, the initial step involves solving an unconstrained optimization problem using stochastic gradient descent (SGD). In the case of a linear model, the optimal solution of this step is typically close to the identity matrix $I$, which is significantly distant from satisfying the directed acyclic graph (DAG) constraints. Using this nearly identity matrix to determine an ordering does not yield any meaningful insights. Furthermore, the next step involves applying SGD to solve a LASSO problem. This is unnecessary since LASSO can be solved much more efficiently using standard methods, rendering the use of SGD redundant in this context. These points suggest that the linear model is not well-suited for this framework.
> >
> > > Local optimal of (11)
> >
> > I do not believe the conclusion from "Universal Gradient Methods for Stochastic Convex Optimization" can be directly applied here to claim that the output corresponds to a local optimal solution of problem (11). The minimum of the ordering does not always equate to a local optimal solution of the loss function (11). This conclusion holds only under certain assumptions, as clearly demonstrated in [4]. To ensure the validity of your theorem, you need to rigorously prove it while explicitly stating and justifying any assumptions required.

---

> ### Author Response · Authors · 2024-11-26
> **Clarifications**
>
> We would like to address the raised concerns.
>
> > _In the case of a linear model, the optimal solution of this step is typically close to the identity matrix I, which is significantly distant from satisfying the directed acyclic graph (DAG) constraints._
>
> We believe there is a misunderstanding. The minimization problem (11) is defined over directed acyclic graphs, and if W represents a valid DAG, all elements on its diagonal are necessarily zero. So, during the whole algorithm, the diagonal of $W_k$ is enforced to be zero. We added these details in the updated paper.  Therefore, the solution of (11) is not close to the identity matrix.
>
> --------
>
> > Theorem 2 & local optimas
>
> You are correct; the solution within a fixed ordering is not necessarily a local solution of (11). We sincerely apologize for this oversight. To address this, we are correcting the claim in Theorem 2 and relegating it to Appendix B.

---

> > ### Comment · Reviewer_pNVL · 2024-11-29
> > **Thanks for the rebuttal**
> >
> > Thank you for addressing my concerns regarding the paper. The revised version is much clearer and more coherent compared to the previous one. Consequently, I am raising my score to 6. However, I believe it would be valuable to explore the extension of the proposed stochastic method to nonlinear models. Since the stochastic algorithm is not inherently restricted to linear or nonlinear models, such an extension could further enhance the paper's contribution.

---

### Official Review · Reviewer_yhkH · 2024-10-28

**Soundness:** 2
**Presentation:** 3
**Contribution:** 2
**Rating:** 3
**Confidence:** 4

**Summary:**

This paper introduces $\psi$DAG, a new method for learning a directed acyclic graph (DAG) from observational data. The new method aims to improve on existing approaches in terms of scalability. Its theoretical complexity is $O(d^2)$, improving on $O(d^3)$ for popular alternatives. This theoretical improvement is demonstrated in computational experiments.

**Strengths:**

The scalability of $\psi$DAG is impressive, and I found the computational experiments convincing. The paper is well-written and contains a good mix of material.

**Weaknesses:**

1. One of the main contributions outlined on page 2 is a proof of Algorithm 1's convergence rate (Theorem 2). I could not see a proof of Theorem 2 in the paper or appendix.
2. The projection algorithm (Algorithm 2) is a key component of the computational machinery, but it remains opaque. It would be insightful to write this projection in terms of an optimization problem and discuss whether the algorithm attains a stationary point of this problem.
3. My primary concern with this paper is the lack of synthetic experiments comparing the *statistical* properties of the competing approaches. Only the computational merits of each approach are compared. It is essential to understand how $\psi$DAG performs on the synthetic data in terms of SHD, TPR, and FPR.
4. A few aspects of the real data example need clarification. First, the reason for having training and testing sets of size $n=853$ and $n=902$ instead of the whole dataset ($n=7466$) is not explained. Second, the role of the testing set is unclear, since the metrics reported do not require a testing set. Finally, I have seen DAGMA run on this dataset before, so I am confused about why it fails now.

**Questions:**

1. Page 1 Line 37: Missing an "and" at the end of the sentence.
2. Page 2 Line 86: Should it say "handling up to 10000 *nodes*"?
3. Page 3 Line 114: Is $X=(X_1,\ldots,X_n)$ a concatenation of column vectors? If so, I think the quantity on the RHS should be transposed. Same for $N$ on Line 126.
4. Page 3: I found it strange that the matrix $W$ is boldface while the matrix $X$ is not.
5. Page 4 Line 168: The $\overset{def}{=}$ looks different here compared with other equations.
6. Page 4 Line 185: The bit that says "where $\xi\in\Xi$ is a random variable drawn from the distribution $\Xi$" confused me. I think it should read "where $\xi$ is a random variable that follows the distribution $\Xi$".
7. Assumption 1: The quantity $\sigma_1^2$ is not defined.
8. Page 4 Line 211: The quantity $R$ is not explained.
9. Page 5 Line 263: $V_J$ should be $V_j$.
10. Figure 1: What is $\bar{x}$?
11. Page 7 Line 330: Can you be more precise than "closest"? Since the projection is heuristic, the word "close" might be better.
12. Page 7 Line 337: What does "topologically through" mean?
13. Figures 2, 3, and 4: The font size is too small for me to read without zooming in.
14. Page 15 Line 799: I am confused about the reference to GPUs in the computing section. Do any of the algorithms use GPUs?
15. Page 17 Line 866: If the edges in the ground truth DAG can be as small as 0.05, why apply a threshold of 0.3? That would make it impossible to ever recover the true graph.
16. Appendix A: I might have missed it, but I could not see the value of $\lambda$ used by $\psi$DAG.

---

> ### Author Response · Authors · 2024-11-19
> **Rebuttal**
>
> We are grateful to the reviewer for the thoughtful review. We are happy to respond to each of the concerns raised.
>
> ----
> >_W1: One of the main contributions outlined on page 2 is a proof of Algorithm 1's convergence rate (Theorem 2). I could not see a proof of Theorem 2 in the paper or appendix._
>
> **Answer:** We would like to apologize for that. The proof of Theorem 2 is a direct application of the optimization algorithm Universal Stochastic Gradient Method (Alg.3 of [SGD]).
> In particular, Theorem 4.2 of [SGD] guarantees that applied to convex L-Hölder continuous functions in the domain of radius $R$, $||x-y|| \leq R$, in $k$ steps decreases functional loss at rate $\mathcal{O} (\frac{LR}{k^2} + \frac {\sigma R}{\sqrt k} )$.
> In our case, once our Algorithm 1 fixes the ordering $\pi_k$ in line 4, DAGs allowed by ordering $\pi_k$ form a linear subspace of $R^{d \times d}$ and in that subspace loss (11) is smooth and convex, and $k$ steps Universal Stochastic Gradient Method converge to a minimum of the ordering at rate given by Theorem 4.2 of [SGD]. This minimum of the ordering is a local minimum of the loss (11).
>
> [SGD] Universal Gradient Methods for Stochastic Convex Optimization, Anton Rodomanov, Ali Kavis, Yongtao Wu, Kimon Antonakopoulos, Volkan Cevher, link:https://openreview.net/pdf?id=Wnhp34K5jR
>
> ----
> >_W2: The projection algorithm (Algorithm 2) is a key component of the computational machinery, but it remains opaque. It would be insightful to write this projection in terms of an optimization problem and discuss whether the algorithm attains a stationary point of this problem._
>
> **Answer:** We would like to point out that the space of all DAGs is a union of numerous linear subspaces, which is extremely nonconvex. Finding the projection (i.e., the closest DAG to a given matrix) is a nontrivial problem that is likely NP-hard. Currently, we do not know how to quantify how close the output of our Alg.2 is to a stationary point of (any) optimization problem.
>
> It is important to note that our framework is not tied to this specific projection method. Any alternative projection method can be seamlessly integrated into our approach.
>
> ___
> >_W3: My primary concern with this paper is the lack of synthetic experiments comparing the statistical properties of the competing approaches. Only the computational merits of each approach are compared. It is essential to understand how $\psi$DAG performs on the synthetic data in terms of SHD, TPR, and FPR._
>
> **Answer:**
> We would like to point out that the statistical properties (SHD, TPR, FPR, etc.) depend on the solution of the optimization problem (11) rather than the optimization algorithms for solving it. In particular, all algorithms minimizing the same optimization problem will output an identical point.
>
> We would like to clarify that the primary contribution of the paper was to establish a flexible and scaleable computational framework for minimizing objective function (11). Adjustments of the objective function to optimize statistical metrics (SHD, TPR, FPR, etc.) were beyond the project's scope and are a direction of our ongoing research.
>
> _____
> >_W4: A few aspects of the real data example need clarification. First, the reason for having training and testing sets of size n =857 and  n=902 instead of the whole dataset (n =7466) is not explained. Second, the role of the testing set is unclear, since the metrics reported do not require a testing set._
>
> **Answer:** We choose only the first subset ($n=853$) to train the models to be consistent with papers [1,2,3,4], which consider the full dataset to be a challenging benchmark (see lines 499-503 in our paper).
> As per the reviewer's request, we added the results for the whole dataset in Appendix B.
>
> > _W4: ...Finally, I have seen DAGMA run on this dataset before, so I am confused about why it fails now._
>
> **Answer:**  We have been puzzled by that as well.
>
> To clarify the situation, the original DAGMA paper does not present a comparison on real datasets. We are aware of only one paper presenting DAGMA evaluation on real datasets, namely [4]. We used the implementation of DAGMA published on their GitHub without any modifications, but we were unable to make it work for neither sample size $n=853$ nor $n=7466$.
>
> ____
> > Regarding questions, typos and presentation suggestions:
>
> **Answer**: We are very grateful to you for helping us improve our paper. Your feedback has been invaluable in polishing our work.

---

> ### Author Response · Authors · 2024-11-19
>
> [1]  Xun Zheng, Bryon Aragam, Pradeep Ravikumar, and Eric Xing. DAGs with no tears: Continuous optimization for structure learning. Advances in Neural Information Processing Systems, 31, 2018.
>
> [2]  Ignavier Ng, AmirEmad Ghassami, and Kun Zhang. On the role of sparsity and DAG constraints for learning linear DAGs. Advances in Neural Information Processing Systems, 33:17943–17954,2020.
>
> [3] Yinghua Gao, Li Shen, and Shu-Tao Xia. DAG-GAN: Causal structure learning with generative adversarial nets. In ICASSP 2021-2021 IEEE International Conference on Acoustics, Speech and Signal Processing (ICASSP), pp. 3320–3324. IEEE, 2021.
>
> [4]  Misiakos, Panagiotis, Chris Wendler, and Markus Püschel. "Learning DAGs from data with few root causes." Advances in Neural Information Processing Systems 36 (2024).

---

> ### Comment · Reviewer_yhkH · 2024-11-25
>
> I thank the authors for their response. However, my main concern regarding the statistical performance of the new approach remains unresolved. In particular, (11) is a nonconvex optimization problem, so the algorithms applied to solve it need not output identical solutions as they do not guarantee a globally optimal solution.
>
> I'm still positive about the scalability of the method and believe the paper is worthy of publication in the future after some additional work.

---

> > ### Author Response · Authors · 2024-11-26
> >
> > We would like to further elaborate on the raised concern.
> >
> > >_In particular, (11) is a nonconvex optimization problem, so the algorithms applied to solve it need not output identical solutions as they do not guarantee a globally optimal solution._
> >
> > It is important to note that the non-convexity of the Directed Acyclic Graph (DAG) constraint presents a significantly greater challenge than the non-convexity of neural network training, where one can often find the global solution.
> >
> > Also, we would like to emphasize that our primary contribution is presenting a framework, not a single specific algorithm. Given the NP-hard nature of the problem, most approaches inevitably rely on meta-heuristics. To address this, we designed our framework to be flexible and compatible with any projection method, and we provided two examples of projection.
> >
> > If we want to make the method more explorative during the projection, we can simply store visited orderings and project to the new one. We can clarify the description of the projection (point 2 in section 4) as follows:
> >
> > 2. "Finding a DAG that is close to the current iterate using a projection $\psi: \mathbb R^{d\times d} \to (\mathbb D, \Pi)$, which also returns its topological sorting $\pi$. For more explorative performance, we recommend saving all previously visited topological orderings $\pi$ and projecting onto a different one in the case of repetition."

---

### Official Review · Reviewer_jVRt · 2024-10-29

**Soundness:** 2
**Presentation:** 3
**Contribution:** 2
**Rating:** 5
**Confidence:** 3

**Summary:**

This paper proposes a novel stochastic approximation framework for learning DAGs, integrating SGD with efficient projection methods to enforce acyclicity. By addressing the non-convexity and complexity challenges of prior approaches, the method achieves low iteration complexity and demonstrates superior scalability and performance in extensive experiments.

**Strengths:**

1. This paper is well-written and easy to follow.

2. This paper introduces a novel stochastic approximation-based method for learning the DAG structure.

3. The paper provides comprehensive numerical experiments, which effectively demonstrate both the performance and efficiency of the proposed methods.

**Weaknesses:**

1. The stochastic approximation approach presented in this paper is limited to linear structural equation models (SEMs). In contrast, other methods such as GOLEM and DAGMA offer extensions to nonlinear SEMs. Expanding the proposed method to nonlinear settings could enhance its contribution.

2. The projection technique introduced in Algorithm 2 appears to lack a clear theoretical guarantee. Moreover, in line 339, the complexity of the projection method is stated as $\mathcal{O}(d^2)$. Given that the numerical experiments involve cases where $d$ can reach $10^4$, the complexity could escalate to approximately 108. This computational burden seems significant and may pose challenges for scalability in practice.

3. The paper could benefit from additional relevant references. For example, Deng et al. [A] propose a topological swap method to accelerate the optimization of the NOTEARS objective by identifying the topological order first and then learning the edge weights of the directed graph. This process aligns with the ideas explored in this paper, and including such a baseline would strengthen the completeness and contextualization of the work.

[A] Deng, Chang, et al. "Optimizing NOTEARS objectives via topological swaps." *International Conference on Machine Learning*. PMLR, 2023.

**Questions:**

1. Can the stochastic approximation method proposed in this paper be extended to nonlinear SEMs? If so, a discussion on the potential challenges or limitations in this direction would be insightful.

2. It appears that Theorem 2 may have been previously developed by Nemirovski and Yudin (1983). Could the authors clarify the novelty of this result, or highlight any differences from the original work?

3. Could the authors elaborate further on the scalability of the proposed method, particularly given the projection method’s complexity?

4. Would the authors consider including the topological swap method from [A] as a baseline? This comparison would provide additional insights into the performance of the proposed method relative to related approaches.

---

> ### Author Response · Authors · 2024-11-19
> **Rebuttal**
>
> We appreciate the reviewer’s constructive feedback and are grateful for the opportunity to address all the raised concerns.
>
> > _Q1: Can the stochastic approximation method proposed in this paper be extended to nonlinear SEMs? If so, a discussion on the potential challenges or limitations in this direction would be insightful_
>
> **Answer:** For simplicity, we have restricted our results to linear SEMs in this paper. However, extending the proposed method to nonlinear SEMs is indeed possible and forms the focus of our ongoing research efforts.
>
> -----
> >_Q2: It appears that Theorem 2 may have been previously developed by Nemirovski and Yudin (1983). Could the authors clarify the novelty of this result, or highlight any differences from the original work?_
>
> **Answer:** We would like to clarify that Theorem 2 quantifies the rate of decrease using an off-the-shelf optimization algorithm Universal Stochastic Gradient Method (Alg. 3 of [SGD]), without any modifications or exploitation of the structure of the considered optimization problem.
>
> [SGD] Universal Gradient Methods for Stochastic Convex Optimization, Anton Rodomanov, Ali Kavis, Yongtao Wu, Kimon Antonakopoulos, Volkan Cevher, link:https://openreview.net/pdf?id=Wnhp34K5jR
>
> ----
> >_The projection technique introduced in Algorithm 2 appears to lack a clear theoretical guarantee. Moreover, in line 339, the complexity of the projection method is stated as  $O(d^2)$. Given that the numerical experiments involve cases where $d$ can reach $10^4$, the complexity could escalate to approximately $10^8$. This computational burden seems significant and may pose challenges for scalability in practice._
>
> >_Q3: Could the authors elaborate further on the scalability of the proposed method, particularly given the projection method’s complexity?_
>
> **Answer:** We would like to clarify that the projection method was not a bottleneck of the algorithm. Observe that the model W itself is a matrix with $d^2$ elements; therefore, for large $d$, $d \approx 10^4$, all direct manipulations of W become computationally challenging. In our simulations, the limiting factor for scalability was not the computation complexity, but the memory requirements to store matrix W itself (see lines 318-319)
>
> -----
> >_Q4: Would the authors consider including the topological swap method from [A] as a baseline? This comparison would provide additional insights into the performance of the proposed method relative to related approaches._
>
> **Answer:** We would like to thank the reviewer for the suggested reference; we are currently adding the method to our experimental comparison. We want to discuss the method in detail.
>
> Reference [A] investigates the NOTEARS framework for continuous optimization in Bayesian network structure learning, providing new theoretical insights into the KKT conditions for such formulations. They propose combining NOTEARS with a local search method informed by KKT conditions (KKTS), leading to improvements in SHD.
>
> While [A] focuses on enhancing existing algorithms with a post-processing step, our work takes an orthogonal approach, reformulating DAG learning as a stochastic optimization problem. We develop an entirely new framework that seamlessly incorporates stochastic gradient methods, allowing efficient processing of large graphs (up to 10,000 nodes), exceeding the computational limits of NOTEARS and [A].

---

> > ### Author Response · Authors · 2024-11-26
> >
> > We would like to thank the reviewer once again for the thoughtful feedback that helped us improve the quality of this submission. If all your comments have been adequately addressed, we humbly request your consideration in reevaluating the score.

---

> > > ### Comment · Reviewer_jVRt · 2024-11-30
> > >
> > > I appreciate the authors' detailed responses. However, I still find the theoretical guarantee of the projection technique in Algorithm 2 unclear. Additionally, the limitation to linear SEM may restrict the broader applicability and impact of this method.

---

### Official Review · Reviewer_DSSv · 2024-10-29

**Soundness:** 2
**Presentation:** 1
**Contribution:** 2
**Rating:** 5
**Confidence:** 2

**Summary:**

This paper studies structure learning in linear DAG models, e.g. to recover the underlying DAG from observational data.
It follows score based learening approach and tries to optimize the square loss. It proposes to apply stochastic gradient descent to solve the optimization problem, and then introduces a procedure to estimate the topological ordering based on learned coefficient matrix.
The experiments are conducted on equal variance noise setting to compare with existing methods.

**Strengths:**

- The proposed method relies on SGD and is fast compared to other continuous optimization approaches for DAG learning.
- Simulation experiments are conducted to show the proposal gives faster convergence speed relative to the optimum.

**Weaknesses:**

- The presentation is poor, many concepts are used without introduction, e.g. transitive closure, full DAG, $R$ in convergence statement.
- In the first step, SGD is applied to solve (11). But actually, the solution to (11) is simply W=I analytically. Also, fixing ordering, the third step  has analytical solution that is simply linear regression of each node onto preceding nodes. All SGD optimizations can be avoided.
- Ordering based DAG learning is a long line of research. This paper lacks citation and discussion on that. Especially, for the equal variance setup considered in this paper, there are scalable algorithms:

[1] Chen, W., Drton, M., & Wang, Y. S. (2019). On causal discovery with an equal-variance assumption. Biometrika, 106(4), 973-980.

[2] Gao, M., Tai, W. M., & Aragam, B. (2022). Optimal estimation of Gaussian DAG models. In International Conference on Artificial Intelligence and Statistics (pp. 8738-8757). PMLR.

**Questions:**

- Where is the proof for Theorem 2? What is the $R$ in the statement? How do we relate this to other optimization results on DAG learning in literature?
- What is the intuition behind Alg 2? There is not explanation why it comes from and why we want to recover the ordering in this way. Any guarantee if it consistently outputs a valid ordering?

---

> ### Author Response · Authors · 2024-11-19
> **Rebuttal part I**
>
> We would like to thank the reviewer for the constructive review. We would like to answer all raised concerns.
>
> >_The presentation is poor, many concepts are used without introduction, e.g. transitive closure, full DAG, R, in convergence statement._
>
> **Answer:** We would like to thank the reviewer for pointing those out, we are correcting them immediately.
>
> -----
> >_In the first step, SGD is applied to solve (11). But actually, the solution to (11) is simply W=I analytically. Also, fixing ordering, the third step has analytical solution that is simply linear regression of each node onto preceding nodes. All SGD optimizations can be avoided._
>
> **Answer:** We believe there may be a misunderstanding. The minimization problem (11) is defined over directed acyclic graphs (DAGs). The identity matrix does not qualify as a DAG. Furthermore, if W represents a valid DAG, all elements on its diagonal are necessarily zero (this is enforced during the whole optimization process) and therefore, the solution of (11) is not close to the identity matrix.
>
> -----
> >_Ordering based DAG learning is a long line of research. This paper lacks citation and discussion on that. Especially, for the equal variance setup considered in this paper, there are scalable algorithms:_
>
> **Answer:** We thank the reviewer for pointing out these relevant algorithms, which have helped us better situate our work within the broader literature.
>
> Reference [1] introduces a procedure for estimating the topological ordering of the underlying graph under the assumption of equal error variances. This approach naturally extends to high-dimensional settings where $p>n$, providing an alternative to greedy search methods with competitive accuracy and computational efficiency. However, unlike our method, it relies on controlling the maximum in-degree of the graph, which can become computationally intensive for graphs with high in-degree.
>
> Our approach, on the other hand, reformulates the DAG learning problem as a stochastic optimization task, leveraging gradient-based methods to scale efficiently to larger graph sizes. While variance-ordering methods usually depends on various error variance assumptions, our framework avoids any such requirements. Additionally, variance-ordering methods can be enhanced with greedy searches to obtain improved performance. Yet, our experimental results show that our method $\psi$DAG achieves state-of-the-art scalability and accuracy without the need for such hybrid extensions. This positions our method as a robust alternative to variance-ordering and greedy approaches, particularly for large-scale DAG learning tasks.
>
> Reference [2] presents a theoretical analysis of Gaussian DAG models, deriving minimax optimal bounds for structure recovery under the equal varience assumption. Their work emphasizes sample efficiency and offers theoretical guarantees for structure learning in Gaussian settings. While this contribution is insightful, it primarily focuses on theoretical aspects and does not address the computational challenges posed by large-scale DAGs. In contrast, our method introduces a scalable stochastic optimization framework that can efficiently handle graphs with up to 10,000 nodes, extending its applicability to a wider range of practical problems.

---

> > ### Author Response · Authors · 2024-11-19
> > **Rebuttal part II**
> >
> > >_Where is the proof for Theorem 2? What is the in the statement? How do we relate this to other optimization results on DAG learning in literature?_
> >
> > **Answer:** We would like to apologize for that. The proof of Theorem 2 is a direct application of the optimization algorithm Universal Stochastic Gradient Method (Alg.3 of [SGD]).
> > In particular, Theorem 4.2 of [SGD] guarantees that applied to convex L-Hölder continuous functions in the domain of radius $R$, $||x-y|| \leq R$, in $k$ steps decreases functional loss at rate $\mathcal{O} \frac{LR}{k^2} + \frac {\sigma R}{\sqrt k} $.
> > In our case, once our Algorithm 1 fixes the ordering $\pi_k$ in line 4, DAGs allowed by ordering $\pi_k$ form a linear subspace of $R^{d \times d}$ and in that subspace loss (11) is smooth and convex, and $k$ steps Universal Stochastic Gradient Method converge to a minimum of the ordering at rate given by Theorem 4.2 of [SGD]. This minimum of the ordering is a local minimum of the loss (11).
> >
> > [SGD] Universal Gradient Methods for Stochastic Convex Optimization, Anton Rodomanov, Ali Kavis, Yongtao Wu, Kimon Antonakopoulos, Volkan Cevher, link:https://openreview.net/pdf?id=Wnhp34K5jR
> >
> > ------
> > >_What is the intuition behind Alg 2? There is not explanation why it comes from and why we want to recover the ordering in this way. Any guarantee if it consistently outputs a valid ordering?_
> >
> > **Answer:** The intuition behind the projection procedure is as follows: In the adjacency matrix of a DAG, the column corresponding to the first vertex in the topological ordering contains all zero elements, and the row corresponding to the last vertex in the topological ordering also contains all zero elements.
> >
> > To project W onto the DAG space, we calculate the norms of all rows and columns. We then identify the smallest norm among them, assigning the corresponding vertex to be either the first or the last in the topological ordering. The remaining vertices are ordered recursively using the same approach.
> >
> > Regarding guarantees for our projection (Algorithm 2), we can guarantee that it'll output a DAG. On the other hand, we do not have a guarantee that the output of projection is the closest DAG (as the space of DAGs is highly nonconvex, this is a nontrivial problem and likely NP-hard).
> >
> > It is important to note that our framework is not tied to this specific projection method. Any alternative projection method can be seamlessly integrated into our approach.

---

> > > ### Author Response · Authors · 2024-11-26
> > >
> > > We would like to thank the reviewer once again for their constructive feedback, which has greatly contributed to improving the quality of our work. If we have successfully addressed all your concerns, we kindly request that you reconsider your score.

---

> > > > ### Comment · Reviewer_DSSv · 2024-11-26
> > > >
> > > > I thank the authors for their response, which partially addressed my concerns. I keep my score unchanged due to the amount of missing details in the current submission and potential revision work.

---

> > > > > ### Author Response · Authors · 2024-11-26
> > > > >
> > > > > Thank you for taking the time to review our paper and for acknowledging our previous response.
> > > > >
> > > > > >_I thank the authors for their response, which partially addressed my concerns._
> > > > >
> > > > > We would kindly ask for clarification regarding which specific concerns remain unaddressed.
> > > > >
> > > > > >_I keep my score unchanged due to the amount of missing details in the current submission and potential revision work._
> > > > >
> > > > > We would like to note that we have updated the submission PDF, highlighting the changes in blue for your convenience. Could you please specify which details might still be missing from the revised version?

---

### Meta-Review · Area_Chair_Bjda · 2024-12-21

**Metareview:**

The paper studies the problem of learning the structure of directed acyclic graphs from data. The main contribution is an optimization based approach for the learning problem that leverages stochastic gradient descent to obtain an efficient and scalable algorithm.

The main strength of this work is that the SGD based algorithm proposed is more efficient and scalable than existing continuous optimization methods for the problem. The reviewers noted several weaknesses and concerns, including that the proposed approach is limited to linear models, some components of the approach lack theoretical justification, the discussion on related work needs to be significantly expanded, and the experimental methodology lacks an evaluation of the statistical performance of the algorithm. Overall, the reviewers appreciated the scalability of the proposed approach but there was consensus that the paper would need additional work and improvements in order for it to be accepted.

**Additional Comments On Reviewer Discussion:**

The reviewers raised several concerns regarding the clarity of the exposition and the discussion of related works. The authors revised the paper in order to incorporate the reviewers' feedback.

---

### Decision · Program_Chairs · 2025-01-22

Reject